# HeadSculpt: Crafting 3D Head Avatars with Text

**Xiao Han**[1,4*] **Yukang Cao**[2*] **Kai Han**[2] **Xiatian Zhu**[1,5]

**Jiankang Deng**[3] **Yi-Zhe Song**[1,4] **Tao Xiang**[1,4†] **Kwan-Yee K. Wong**[2†]

[1]University of Surrey    [2]The University of Hong Kong    [3]Imperial College London
[4]iFlyTek-Surrey Joint Research Centre on AI    [5]Surrey Institute for People-Centred AI

## Abstract

Recently, text-guided 3D generative methods have made remarkable advancements in producing high-quality textures and geometry, capitalizing on the proliferation of large vision-language and image diffusion models. However, existing methods still struggle to create high-fidelity 3D head avatars in two aspects: (1) They rely mostly on a pre-trained text-to-image diffusion model whilst missing the necessary 3D awareness and head priors. This makes them prone to inconsistency and geometric distortions in the generated avatars. (2) They fall short in fine-grained editing. This is primarily due to the inherited limitations from the pre-trained 2D image diffusion models, which become more pronounced when it comes to 3D head avatars. In this work, we address these challenges by introducing a versatile coarse-to-fine pipeline dubbed **HeadSculpt** for crafting (*i.e.*, generating and editing) 3D head avatars from textual prompts. Specifically, we first equip the diffusion model with 3D awareness by leveraging landmark-based control and a learned textual embedding representing the back view appearance of heads, enabling 3D-consistent head avatar generations. We further propose a novel identity-aware editing score distillation strategy to optimize a textured mesh with a high-resolution differentiable rendering technique. This enables identity preservation while following the editing instruction. We showcase HeadSculpt's superior fidelity and editing capabilities through comprehensive experiments and comparisons with existing methods. [‡]

## 1 Introduction

Modeling 3D head avatars underpins a wide range of emerging applications (*e.g.*, digital telepresence, game character creation, and AR/VR). Historically, the creation of intricate and detailed 3D head avatars demanded considerable time and expertise in art and engineering. With the advent of deep learning, existing works [87, 28, 33, 72, 8, 38, 15] have shown promising results on the reconstruction of 3D human heads from monocular images or videos. However, these methods remain restricted to head appearance contained in their training data which is often limited in size, resulting in the inability to generalize to new appearance beyond the training data. This constraint calls for the need of more flexible and generalizable methods for 3D head modeling.

Recently, vision-language models (*e.g.*, CLIP [55]) and diffusion models (*e.g.*, Stable Diffusion [69, 61, 59]) have attracted increasing interest. These progresses have led to the emergence of text-to-3D generative models [34, 62, 44, 27] which create 3D content in a self-supervised manner. Notably, DreamFusion [54] introduces a score distillation sampling (SDS) strategy that leverages a pre-trained image diffusion model to compute the noise-level loss from the textual description, unlocking the potential to optimize differentiable 3D scenes (*e.g.*, neural radiance field [45], tetrahedral mesh [66], texture [58, 9], or point cloud [50]) with 2D diffusion prior only. Subsequent research efforts [43, 6,

---

*Equal contributions    † Corresponding authors    ‡ Webpage: https://brandonhan.uk/HeadSculpt

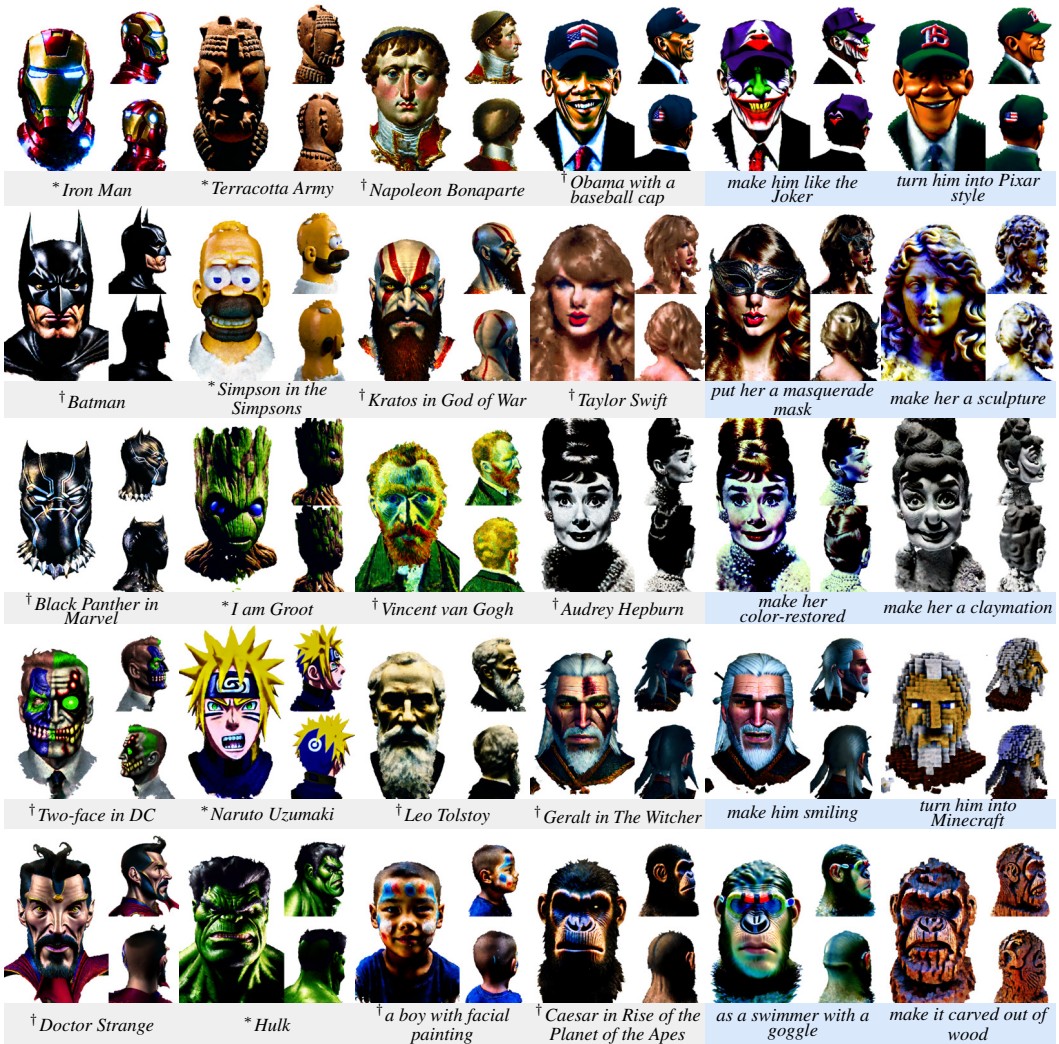

Figure 1: **Examples of generation and editing results obtained using the proposed HeadSculpt.** It enables the creation and fine-grained editing of high-quality head avatars, featuring intricate geometry and texture, for any type of head avatar using simple descriptions or instructions. Symbols indicate the following prompt prefixes: ∗ "a head of [text]" and † "a DSLR portrait of [text]". The captions in gray are the prompt suffixes while the blue ones are the editing instructions.

65, 79, 42, 75, 40, 56, 76] improve and extend DreamFusion from various perspectives (*e.g.*, higher resolution [39] and better geometry [10]).

Considering the flexibility and versatility of natural languages, one might think that these SDS-based text-to-3D generative methods would be sufficient for generating diverse 3D avatars. However, it is noted that existing methods have two major drawbacks (see Fig. 6): *(1) Inconsistency and geometric distortions*: The 2D diffusion models used in these methods lack 3D awareness particularly regarding camera pose; without any remedy, existing text-to-3D methods inherited this limitation, leading to the multi-face *"Janus"* problem in the generated head avatars. *(2) Fine-grained editing limitations*: Although previous methods propose to edit 3D models by naively fine-tuning trained models with modified prompts [54, 39], we find that this approach is prone to biased outcomes, such as identity loss or inadequate editing. This problem arises from two causes: (a) inherent bias in prompt-based editing in image diffusion models, and (b) challenges with inconsistent gradient back-propagation at separate iterations when using SDS calculated from a vanilla image diffusion model.

In this paper, we introduce a new head-avatar-focused text-to-3D method, dubbed **HeadSculpt**, that supports high-fidelity generation and fine-grained editing. Our method comprises two novel

components: *(1) Prior-driven score distillation:* We first arm the pre-trained image diffusion model with 3D awareness by integrating a landmark-based ControlNet [84]. Specifically, we adopt the parametric 3D head model, FLAME [38], as a prior to obtain a 2D landmark map [41, 31], which serves as an additional condition for the diffusion model, ensuring the consistency of generated head avatars across different views. Further, to remedy the front-view bias in the pre-trained diffusion model, we utilize an improved view-dependent prompt through textual inversion [17], by learning a specialized `<back-view>` token to emphasize back views of heads and capture their unique visual details. *(2) Identity-aware editing score distillation (IESD):* To address the challenges of fine-grained editing for head avatars, we introduce a novel method called IESD. It blends two scores, one for editing and the other for identity preservation, both predicted by a ControlNet-based implementation of InstructPix2Pix [5]. This approach maintains a controlled editing direction that respects both the original identity and the editing instructions. To further improve the fidelity of our method, we integrate these two novel components into a coarse-to-fine pipeline [39], utilizing NeRF [48] as the low-resolution coarse model and DMTET [66] as the high-resolution fine model. As demonstrated in Fig. 1, our method can generate high-fidelity human-like and non-human-like head avatars while enabling fine-grained editing, including local changes, shape/texture modifications, and style transfers.

## 2   Related work

**Text-to-2D generation.** In recent years, groundbreaking vision-language technologies such as CLIP [55] and diffusion models [25, 13, 59, 68] have led to significant advancements in text-to-2D content generation [61, 57, 1, 69, 70]. Trained on extensive 2D multimodal datasets [63, 64], they are empowered with the capability to *"dream"* from the prompt. Follow-up works endeavor to efficiently control the generated results [84, 85, 47], extend the diffusion model to video sequence [67, 3], accomplish image or video editing [23, 32, 81, 5, 77, 14, 22], enhance the performance for personalized subjects [60, 17], etc. Although significant progress has been made in generating 2D content from text, carefully crafting the prompt is crucial, and obtaining the desired outcome often requires multiple attempts. The inherent randomness remains a challenge, especially for editing tasks.

**Text-to-3D generation.** Advancements in text-to-2D generation have paved the way for text-to-3D techniques. Early efforts [82, 27, 44, 62, 34, 29, 11] propose to optimize the 3D neural radiance field (NeRF) or vertex-based meshes by employing the CLIP language model. However, these models encounter difficulties in generating expressive 3D content, primarily because of the limitations of CLIP in comprehending natural language. Fortunately, the development of image diffusion models [69, 1] has led to the emergence of DreamFusion [54]. It proposes Score Distillation Sampling (SDS) based on a pre-trained 2D diffusion prior [61], showcasing promising generation results. Subsequent works [37] have endeavored to improve DreamFusion from various aspects: Magic3D [39] proposes a coarse-to-fine pipeline for high-resolution generations; Latent-NeRF [43] includes shape guidance for more robust generation on the latent space [59]; DreamAvatar [6] leverages SMPL [4] to generate 3D human full-body avatars under controllable shapes and poses; Guide3D [7] explores the usage of multi-view generated images to create 3D human avatars; Fantasia3D [10] disentangles the geometry and texture training with DMTET [66] and PBR texture [49] as their 3D representation; 3DFuse [65] integrates depth control and semantic code sampling to stabilize the generation process. Despite notable progress, current text-to-3D generative models still face challenges in producing view-consistent 3D content, especially for intricate head avatars. This is primarily due to the absence of 3D awareness in text-to-2D diffusion models. Additionally, to the best of our knowledge, there is currently no approach that specifically focuses on editing the generated 3D content, especially addressing the intricate fine-grained editing needs of head avatars.

**3D head modeling and creation.** Statistical mesh-based models, such as FLAME [38, 15], enable the reconstruction of 3D head models from images. However, they struggle to capture fine details like hair and wrinkles. To overcome this issue, recent approaches [8, 71, 72, 51] employ Generative Adversarial Networks (GANs) [46, 20, 30] to train 3D-aware networks on 2D head datasets and produce 3D-consistent images through latent code manipulation. Furthermore, neural implicit methods [87, 16, 28, 88] introduce implicit and subject-oriented head models based on neural rendering fields [45, 48, 2]. Recently, text-to-3D generative methods have gained traction, generating high-quality 3D head avatars from natural language using vision-language models [55, 69]. Typically, T2P [85] predicts bone-driven parameters of head avatars via a game engine under the CLIP guidance [55]. Rodin [80] proposes a roll-out diffusion network to perform 3D-aware diffusion. DreamFace [83] employs a

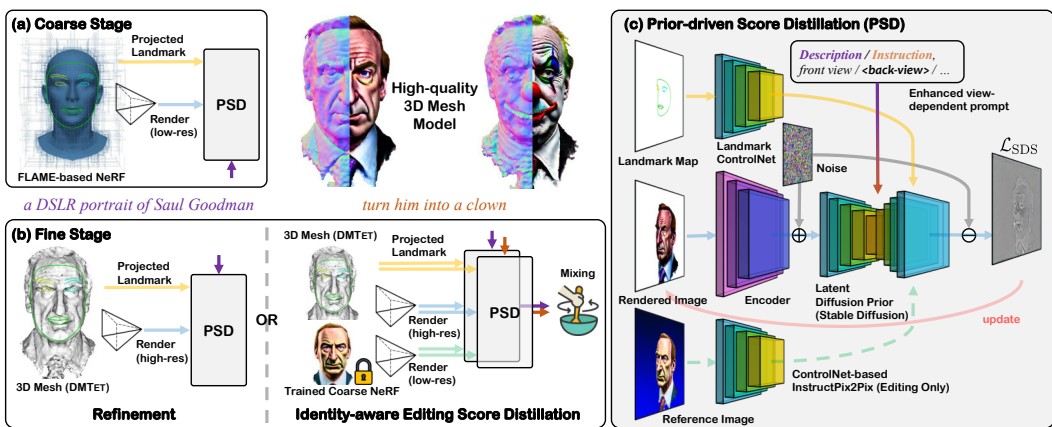

Figure 2: **Overall architecture of HeadSculpt.** We craft high-resolution 3D head avatars in a coarse-to-fine manner. **(a)** We optimize neural field representations for the coarse model. **(b)** We refine or edit the model using the extracted 3D mesh and apply identity-aware editing score distillation if editing is the target. **(c)** The core of our pipeline is the prior-driven score distillation, which incorporates landmark control, enhanced view-dependent prompts, and an InstructPix2Pix branch.

selection strategy in the CLIP embedding space to generate coarse geometry and uses SDS [54] to optimize UV-texture. Despite producing promising results, all these methods require a large amount of data for supervised training and struggle to generalize well to non-human-like avatars. In contrast, our approach relies solely on pre-trained text-to-2D models, generalizes well to out-of-domain avatars, and is capable of performing fine-grained editing tasks.

## 3 Methodology

HeadSculpt is a 3D-aware text-to-3D approach that utilizes a pre-trained text-to-2D Stable Diffusion model [69, 59] to generate high-resolution head avatars and perform fine-grained editing tasks. As illustrated in Fig. 2, the generation pipeline has two stages: coarse generation via the neural radiance field (NeRF) [48] and refinement/editing using tetrahedron mesh (DMTET) [66]. Next, we will first introduce the preliminaries that form the basis of our method in Sec. 3.1. We will then discuss the key components of our approach in Sec. 3.2 and Sec. 3.3, including (1) the prior-driven score distillation process via landmark-based ControlNet [84] and textual inversion [17], and (2) identity-aware editing score distillation accomplished in the fine stage using the ControlNet-based InstructPix2Pix [5].

### 3.1 Preliminaries

**Score distillation sampling.** Recently, DreamFusion [54] proposed score distillation sampling (SDS) to self-optimize a text-consistent neural radiance field (NeRF) based a the pre-trained text-to-2D diffusion model [61]. Due to the unavailability of the Imagen model [61] used by DreamFusion, we employ the latent diffusion model in [59] instead. Specifically, given a latent feature $z$ encoded from an image $x$, SDS introduces random noise $\epsilon$ to $z$ to create a noisy latent variable $z_t$ and then uses a pre-trained denoising function $\epsilon_\phi(z_t; y, t)$ to predict the added noise. The SDS loss is defined as the difference between predicted and added noise and its gradient is given by

$$\nabla_\theta \mathcal{L}_{\text{SDS}}(\phi, g(\theta)) = \mathbb{E}_{t, \epsilon \sim \mathcal{N}(0,1)} \left[ w(t) \left( \epsilon_\phi(z_t; y, t) - \epsilon \right) \frac{\partial z}{\partial x} \frac{\partial x}{\partial \theta} \right], \tag{1}$$

where $y$ is the text embedding, $w(t)$ weights the loss from noise level $t$. With the expressive text-to-2D diffusion model and self-supervised SDS loss, we can back-propagate the gradients to optimize an implicit 3D scene $g(\theta)$, eliminating the need for an expensive 3D dataset.

**3D scene optimization.** HeadSculpt explores the potential of two different 3D differentiable representations as the optimization basis for crafting 3D head avatars. Specifically, we employ NeRF [48] in the coarse stage due to its greater flexibility in geometry deformation, while utilizing DMTET [66] in the fine stage for efficient high-resolution optimization.

**(1)** *3D prior-based NeRF.* DreamAvatar [6] recently proposed a density-residual setup to enhance the robustness and controllability of the generated 3D NeRF. Given a point $\mathbf{x}$ inside the 3D volume, we can derive its density and color value based on a prior-based density field $\bar{\sigma}$:

$$F(\mathbf{x}, \bar{\sigma}) = F_\theta(\gamma(\mathbf{x})) + (\bar{\sigma}(\mathbf{x}), \mathbf{0}) \mapsto (\sigma, \mathbf{c}), \tag{2}$$

where $\gamma(\cdot)$ denotes a hash-grid frequency encoder [48], and $\sigma$ and $\mathbf{c}$ are the density and RGB color respectively. We can derive $\bar{\sigma}$ from the signed distance $d(\mathbf{x})$ of a given 3D shape prior (*e.g.*, a canonical FLAME model [38] by default in our implementation):

$$\bar{\sigma}(\mathbf{x}) = \max\left(0, \text{softplus}^{-1}(\tau(\mathbf{x}))\right), \ \tau(\mathbf{x}) = \frac{1}{a}\text{sigmoid}(-d(\mathbf{x})/a), \text{where } a = 0.005. \tag{3}$$

To obtain a 2D RGB image from the implicit volume defined above, we employ a volume rendering technique that involves casting a ray $\mathbf{r}$ from the 2D pixel location into the 3D scene, sampling points $\boldsymbol{\mu}_i$ along the ray, and calculating their density and color value using $F$ in Eq. (2):

$$C(\mathbf{r}) = \sum_i W_i \mathbf{c}_i, \quad W_i = \alpha_i \prod_{j<i}(1 - \alpha_j), \quad \alpha_i = 1 - e^{(-\sigma_i||\boldsymbol{\mu}_i - \boldsymbol{\mu}_{i+1}||)}. \tag{4}$$

**(2) DMTET.** It discretizes a deformable tetrahedral grid $(V_T, T)$, where $V_T$ denotes the vertices within grid $T$ [19, 66], to model the 3D space. Every vertex $\mathbf{v}_i \in V_T \subset \mathbb{R}^3$ possesses a signed distance value $s_i \in \mathbb{R}$, along with a position offset $\Delta\mathbf{v}_i \in \mathbb{R}^3$ of the vertex relative to its initial canonical coordinates. Subsequently, the underlying mesh can be extracted based on $s_i$ with the differentiable marching tetrahedra algorithm. In addition to the geometry, we adopt the Magic3D approach [39] to construct a neural color field. This involves re-utilizing the MLP trained in the coarse NeRF stage to predict the RGB color value for each 3D point. During optimization, we render this textured surface mesh into high-resolution images using a differentiable rasterizer [36, 49].

## 3.2 3D-Prior-driven score distillation

Existing text-to-3D methods with SDS [54] assume that maximizing the likelihood of images rendered from various viewpoints of a scene model $g(\cdot)$ is equivalent to maximizing the overall likelihood of $g(\cdot)$. This assumption can result in inconsistencies and geometric distortions [54, 65]. A notable issue is the "*Janus problem*" characterized by multiple faces on a single object (see Fig. 6). There are two possible causes: (1) the randomness of the diffusion model which can cause inconsistencies among different views, and (2) the lack of 3D awareness in controlling the generation process, causing the model to struggle in determining the front view, back view, etc. To address these issues in generating head avatars, we integrate 3D head priors into the diffusion model.

**Landmark-based ControlNet.** In Section 3.1, we explain our adoption of FLAME [38] as the density guidance for our NeRF. Nevertheless, this guidance by itself is insufficient to have a direct impact on the SDS loss. What is missing is a link between the NeRF and the diffusion model, incorporating the same head priors. Such a link is key to improving the view consistency of the generated head avatars. To achieve this objective, as illustrated in Fig. 2, we propose the incorporation of 2D landmark maps as an additional condition for the diffusion model using ControlNet [84]. Specifically, we employ a ControlNet $\mathcal{C}$ trained on a large-scale 2D face dataset [86, 12], using facial landmarks rendered from MediaPipe [41, 31] as ground-truth data. When given a randomly sampled camera pose $\pi$, we first project the vertices of the FLAME model onto the image. Following that, we select and render some of these vertices into a landmark map $\mathcal{P}_\pi$ based on some predefined vertex indexes. The landmark map will be fed into ControlNet and its output features are added to the intermediate features within the diffusion U-Net. The gradient of our SDS loss can be re-written as

$$\nabla_\theta \mathcal{L}_{\text{SDS}}(\phi, g(\theta)) = \mathbb{E}_{t,\epsilon\sim\mathcal{N}(0,1),\pi}\left[w(t)\left(\epsilon_\phi\left(z_t; y, t, \mathcal{C}(\mathcal{P}_\pi)\right) - \epsilon\right)\frac{\partial z}{\partial x}\frac{\partial x}{\partial \theta}\right]. \tag{5}$$

**Enhanced view-dependent prompt via textual inversion.** Although the landmark-based ControlNet can inject 3D awareness into the pre-trained diffusion models, it struggles to maintain back view head consistency. This is expected as the 2D image dataset used for training mostly contains only front or side face views. Consequently, when applied directly to back views, the model introduces ambiguity as front and back 3D landmark views can appear similar, as shown in Fig. 8. To address this issue, we propose a simple yet effective method. Our method is inspired by previous works [54, 65, 39] which found it beneficial to append view-dependent text (*e.g.*, "front view", "side view" or "back

view") to the provided input text based on the azimuth angle of the randomly sampled camera. We extend this idea by learning a special token `<back-view>` to replace the plain text "back view" in order to emphasize the rear appearance of heads. This is based on the assumption that a pre-trained Stable Diffusion does has the ability to "imagine" the back view of a head - it has seen some during training. The main problem is that a generic text embedding of "back view" is inadequate in telling the model what appearance it entails. A better embedding for "back view" is thus required. To this end, we first randomly download 34 images of the back view of human heads, without revealing any personal identities, to construct a tiny dataset $\mathcal{D}$, and then we optimize the special token $v$ (*i.e.*, `<back-view>`) to better fit the collected images, similar to the textual inversion [17]:

$$v_* = \arg\min_v \mathbb{E}_{t,\epsilon \sim \mathcal{N}(0,1), z \sim \mathcal{D}} \left[ \|\epsilon - \epsilon_\phi (z_t; v, t)\|_2^2 \right], \tag{6}$$

which is achieved by employing the same training scheme as the original diffusion model, while keeping $\epsilon_\phi$ fixed. This constitutes a reconstruction task, which we anticipate will encourage the learned embedding to capture the fine visual details of the back views of human heads. Notably, as we do not update the weights of $\epsilon_\phi$, it stays compatible with the landmark-based ControlNet.

### 3.3  Identity-aware editing score distillation

After generating avatars, editing them to fulfill particular requirements poses an additional challenge. Previous works [54, 39] have shown promising editing results by fine-tuning a trained scene model with a new target prompt. However, when applied to head avatars, these methods often suffer from identity loss or inadequate appearance modifications (see Fig. 10). This problem stems from the inherent constraint of the SDS loss, where the 3D models often sacrifice prominent features to preserve view consistency. Substituting Stable Diffusion with InstructPix2Pix [5, 21] might seem like a simple solution, but it also faces difficulties in maintaining facial identity during editing based only on instructions, as it lacks a well-defined anchor point.

To this end, we propose identity-aware editing score distillation (IESD) to regulate the editing direction by blending two predicted scores, *i.e.*, one for editing instruction and another for the original description. Rather than using the original InstructPix2Pix [5], we employ a ControlNet-based InstructPix2Pix $\mathcal{I}$ [84] trained on the same dataset, ensuring compatibility with our landmark-based ControlNet $\mathcal{C}$ and the learned `<back-view>` token. Formally, given an initial textual prompt $y$ describing the avatar to be edited and an editing instruction $\hat{y}$, we first input them separately into the same diffusion model equipped with two ControlNets, $\mathcal{I}$ and $\mathcal{C}$. This allows us to obtain two predicted noises, which are then combined using a predefined hyper-parameter $\omega_e$ like classifier-free diffusion guidance (CFG) [26]:

$$\nabla_\theta \mathcal{L}_{\text{IESD}}(\phi, g(\theta)) = \mathbb{E}_{t,\epsilon \sim \mathcal{N}(0,1), \pi} \left[ w(t) \left( \underbrace{\hat{\epsilon}_\phi (z_t; y, \hat{y}, t, \mathcal{C}(\mathcal{P}_\pi), \mathcal{I}(\mathcal{M}_\pi))} - \epsilon \right) \frac{\partial z}{\partial x} \frac{\partial x}{\partial \theta} \right], \tag{7}$$
$$\omega_e \epsilon_\phi (z_t; \hat{y}, t, \mathcal{C}(\mathcal{P}_\pi), \mathcal{I}(\mathcal{M}_\pi)) + (1 - \omega_e) \epsilon_\phi (z_t; y, t, \mathcal{C}(\mathcal{P}_\pi), \mathcal{I}(\mathcal{M}_\pi))$$

where $\mathcal{P}_\pi$ and $\mathcal{M}_\pi$ represent the 2D landmark maps and the reference images rendered in the coarse stage, both being obtained under the sampled camera pose $\pi$. The parameter $\omega_e$ governs a trade-off between the original appearance and the desired editing, which defaults to 0.6 in our experiments.

## 4  Experiments

We will now assess the efficacy of our HeadSculpt across different scenarios, while also conducting a comparative analysis against state-of-the-art text-to-3D generation pipelines.

**Implementation details.** HeadSculpt builds upon Stable-DreamFusion [73] and Huggingface Diffusers [78, 53]. We utilize version 1.5 of Stable Diffusion [69] and version 1.1 of ControlNet [84, 12] in our implementation. In the coarse stage, we optimize our 3D model at $64 \times 64$ grid resolution, while using $512 \times 512$ grid resolution for the fine stage (refinement or editing). Typically, each text prompt requires approximately $7,000$ iterations for the coarse stage and $5,000$ iterations for the fine stage. It takes around 1 hour for each stage on a single Tesla V100 GPU with a default batch size of 4. We use Adam [35] optimizer with a fixed learning rate of $0.001$. Additional implementation details can be found in the supplementary material.

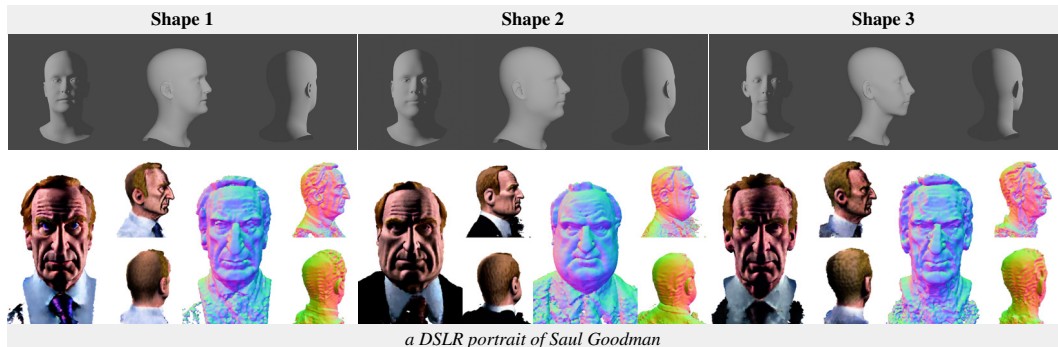

Figure 3: **Generation results with various shapes.** The first row shows three randomly sampled FLAME models, while the second row presents our generated results (incl. normals) using these FLAME models as initialization. All results are under the same text prompt.

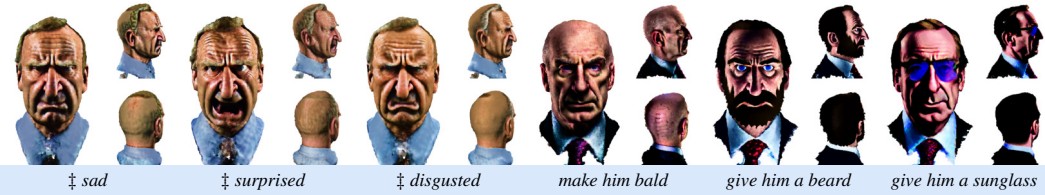

Figure 4: **More specific editing results.** ‡ Instruction prefix: *make his expression as* [text].

**Baseline methods for generation evaluation.** We compare the generation results with five baselines: DreamFusion [73], Latent-NeRF [43], 3DFuse [65] (improved version of SJC [79]), Fantasia3D [10], and DreamFace [83]. We do not directly compare with DreamAvatar [6] as it involves deformation fields for full-body-related tasks.

**Baseline methods for editing evaluation.** We assess IESD's efficacy for fine-grained 3D head avatar editing by comparing it with various alternatives since no dedicated method exists for this: **(B1)** One-step optimization on the coarse stage without initialization; **(B2)** Initialized from the coarse stage, followed by optimization of another coarse stage with an altered description; **(B3)** Initialized from the coarse stage, followed by optimization of a new fine stage with an altered description; **(B4)** Initialized from the coarse stage, followed by optimization of a new fine stage with an instruction based on the vanilla InstructPix2Pix [5]; **(B5)** Ours without edit scale (*i.e.*, $\omega_e = 1$). Notably, B2 represents the editing method proposed in DreamFusion [54], while B3 has a similar performance as Magic3D [39], which employs a three-stage editing process (*i.e.*, Coarse + Coarse + Fine).

## 4.1 Qualitative evaluations

**Head avatar generation with various prompts.** In Fig. 1, we show a diverse array of 3D head avatars generated by our HeadSculpt, consistently demonstrating high-quality geometry and texture across various viewpoints. Our method's versatility is emphasized by its ability to create an assortment of avatars, including humans (both celebrities and ordinary individuals) as well as non-human characters like superheroes, comic/game characters, paintings, and more.

**Head avatar generation with different shapes.** HeadSculpt leverages shape-guided NeRF initialization and landmark-guided diffusion priors. This allows controlling geometry by varying the FLAME shape used for initialization. To demonstrate adjustability, Fig. 3 presents examples generated from diverse FLAME shapes. The results fit closely to the shape guidance, highlighting HeadSculpt's capacity for geometric variation when provided different initial shapes.

**Head avatar editing with various instructions.** As illustrated in Fig. 1 and Fig. 4, HeadSculpt's adaptability is also showcased through its ability to perform fine-grained editing, such as local changes (*e.g.*, adding accessories, changing hairstyles, or altering expressions), shape and texture modifications, and style transfers.

**Head avatar editing with different edit scales.** In Fig. 5, we demonstrate the effectiveness of IESD with different $\omega_e$ values, highlighting its ability to control editing influence on the reference identity.

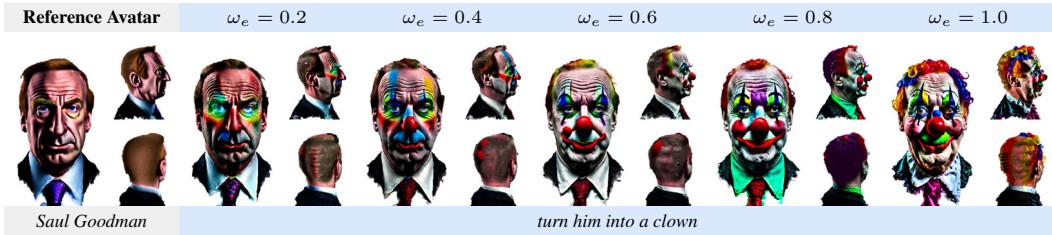

Figure 5: **Impact of the edit scale** $\omega_e$ **in IESD.** It balances the preservation of the initial appearance and the extent of the desired editing, making the editing process more controllable and flexible.

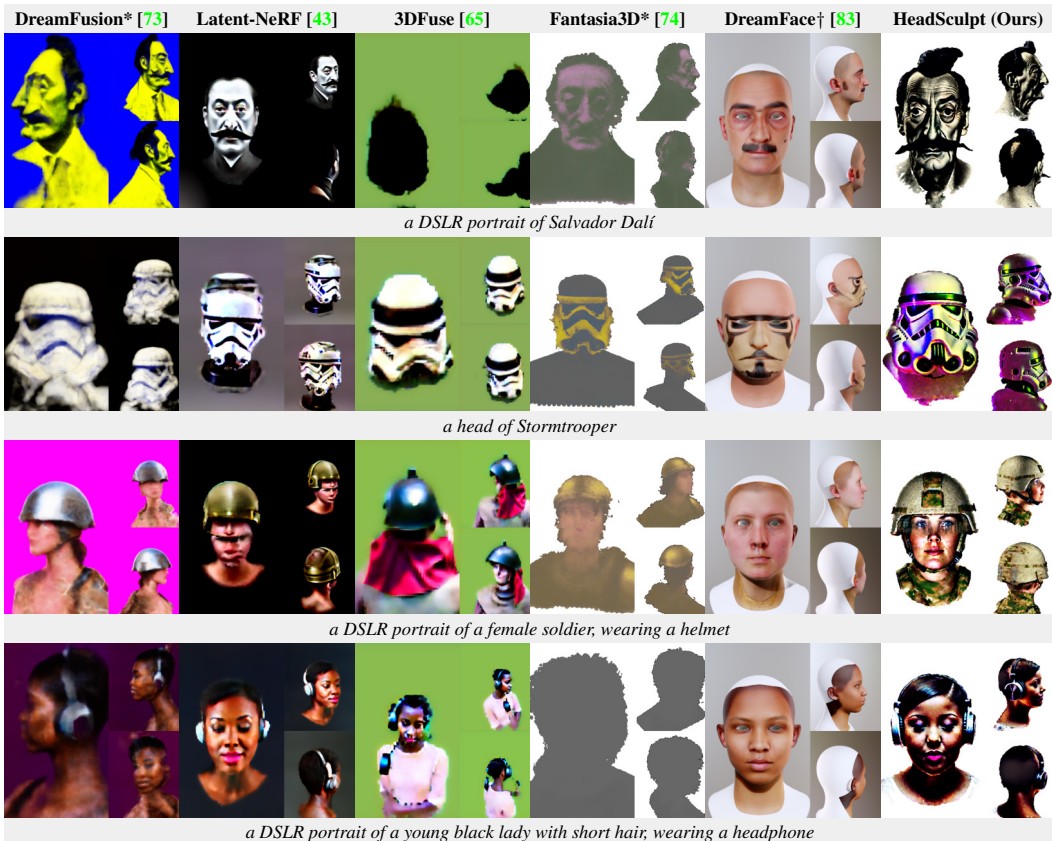

Figure 6: **Comparison with existing text-to-3D methods.** Unlike other methods that struggle or fail to generate reasonable results, our approach consistently achieves high-quality geometry and texture, yielding superior results. *Non-official implementation. † Generated from the online website demo.

**Comparison with existing methods on generation results.** We provide qualitative comparisons with existing methods in Fig. 6. We employ the same FLAME model for Latent-NeRF [43] to compute their sketch-guided loss and for Fantasia3D [74] as the initial geometry. The following observations can be made: **(1)** All baselines tend to be more unstable during training than ours, often resulting in diverged training processes; **(2)** Latent-NeRF occasionally produces plausible results due to its use of the shape prior, but its textures are inferior to ours since optimization occurs solely in the latent space; **(3)** Despite 3DFuse's depth control to mitigate the Janus problem, it still struggles to generate 3D consistent head avatars; **(4)** While Fantasia3D can generate a mesh-based 3D avatar, its geometry is heavily distorted, as its disentangled geometry optimization might be insufficient for highly detailed head avatars; **(5)** Although DreamFace generates realistic human face textures, it falls short in generating (i) complete heads, (ii) intricate geometry, (iii) non-human-like appearance, and (iv) composite accessories. In comparison, our method consistently yields superior results in both geometry and texture with much better consistency for the given prompt. More comparisons can be found in the supplementary material.

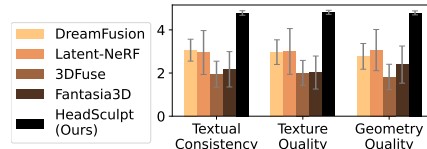

| Generation | Dream Fusion | Latent-NeRF | 3DFuse | Fantasia3D | Ours |
|---|---|---|---|---|---|
| **CLIP-R** [52] | 95.83 | 87.50 | 70.83 | 62.50 | **100.00** |
| **CLIP-S** [24] | 26.06 | 26.30 | 23.41 | 23.26 | **29.52** |
| **Editing** | | **B3** | **B4** | **B5** | **Ours** |
| **CLIP-DS** [18] | | 16.62 | 8.76 | 14.03 | **16.84** |

Figure 7: **User study.** Numbers are averaged over 42 responses.

Table 1: **Objective evaluation with CLIP-based metrics.** All numbers are calculated with CLIP-L/14.

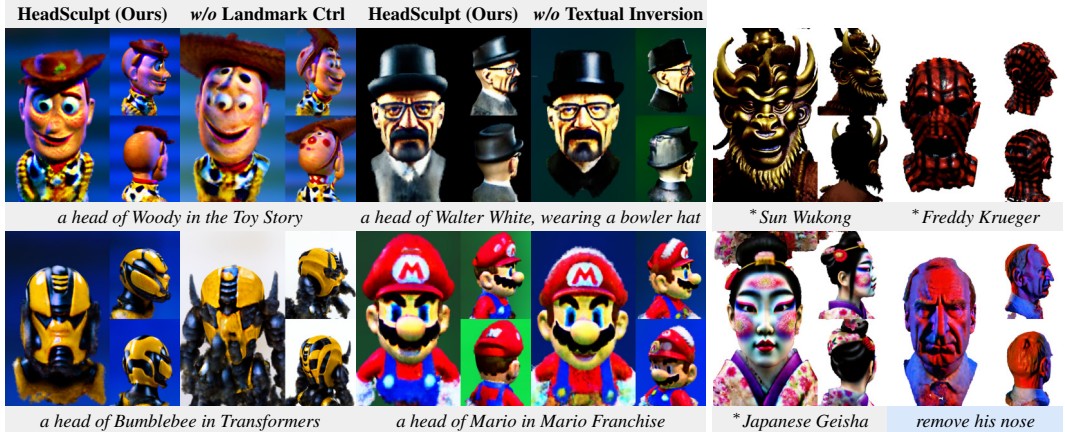

Figure 8: **Analysis of prior-driven score distillation.**

Figure 9: **Failure cases.**

## 4.2 Quantitative evaluations

**User studies.** We conducted user studies comparing with four baselines [73, 74, 65, 43]. 42 volunteers ranked them from 1 (worst) to 5 (best) individually based on three dimensions: **(1)** consistency with the text, **(2)** texture quality, and **(3)** geometry quality. The results, shown in Fig. 7, indicate that our method achieved the highest rank in all three aspects by large margins.

**CLIP-based metrics.** Following DreamFusion [54], **(1)** We calculate the CLIP R-Precision (CLIP-R) [52] and CLIP-Score (CLIP-S) [24] metrics, which evaluate the correlation between the generated images and the input texts, for all methods using 30 text prompts. As indicated in Tab. 1, our approach significantly outperforms the competing methods according to both metrics. This outcome provides additional evidence for the subjective superiority observed in the user studies and qualitative results. **(2)** We employ the CLIP Directional Similarity (CLIP-DS) [5, 18], to evaluate the editing performance. This metric measures the alignment between changes in text captions and corresponding image modifications. Specifically, we encode a pair of images (the original and edited 3D models rendered from a specific viewpoint) along with a pair of text prompts describing the original and edited subjects, *e.g.*, "a DSLR portrait of Saul Goodman" and "a DSLR portrait of Saul Goodman dressed like a clown". We compare our approach against B3, B4, and B5 by evaluating 10 edited examples. The results, presented in Tab. 1, highlight the superiority of our editing framework according to this metric, indicating improved editing fidelity and identity preservation compared to alternatives.

## 4.3 Further analysis

**Effectiveness of prior-driven score distillation.** In Fig. 8, we conduct ablation studies to examine the impact of the proposed landmark control and textual inversion priors in our method. We demonstrate this on the coarse stage because the refinement and editing results heavily depend on this stage. The findings show that landmark control is essential for generating spatially consistent head avatars. Without it, the optimized 3D avatar faces challenges in maintaining consistent facial views, particularly for non-human-like characters. Moreover, textual inversion is shown to be another vital component in mitigating the Janus problem, specifically for the back view, as landmarks cannot exert control on the rear view. Overall, the combination of both components enables HeadSculpt to produce view-consistent avatars with high-quality geometry.

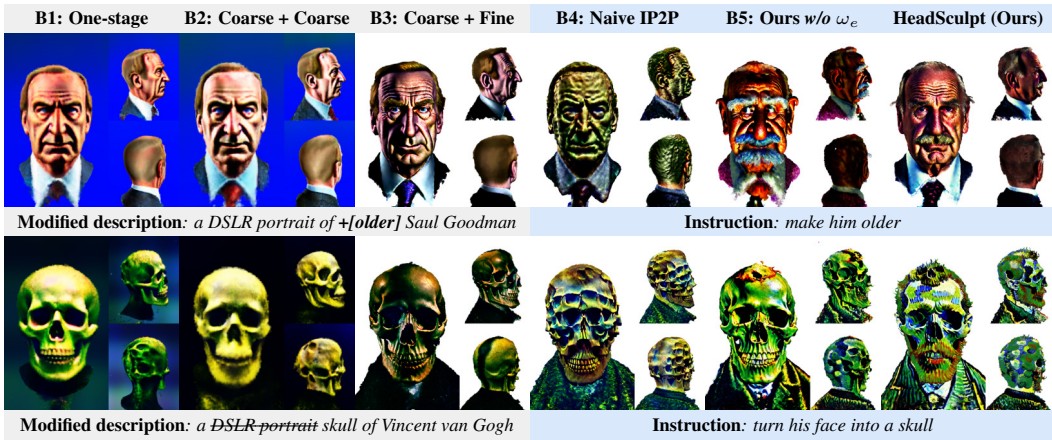

| B1: One-stage | B2: Coarse + Coarse | B3: Coarse + Fine | B4: Naive IP2P | B5: Ours *w/o* $\omega_e$ | HeadSculpt (Ours) |

**Modified description**: *a DSLR portrait of +[older] Saul Goodman*     **Instruction**: *make him older*

**Modified description**: *a ~~DSLR portrait~~ skull of Vincent van Gogh*     **Instruction**: *turn his face into a skull*

Figure 10: **Analysis of identity-aware editing score distillation.**

**Effectiveness of IESD.** In Fig. 10, we present two common biased editing scenarios produced by the baseline methods: insufficient editing and loss of identity. With Stable Diffusion, specific terms like "Saul Goodman" and "skull" exert a more substantial influence on the text embeddings compared to other terms, such as "older" and "Vincent van Gogh". B1, B2, and B3, all based on vanilla Stable Diffusion, inherit such bias in their generated 3D avatars. Although B4 does not show such bias, it faces two other issues: **(1)** the Janus problem reemerges due to incompatibility between vanilla InstructPix2Pix and the proposed prior-driven score distillation; **(2)** it struggles to maintain facial identity during editing based solely on instructions, lacking a well-defined anchor point. In contrast, B5 employs ControlNet-based InstructPix2Pix [84] with the proposed prior score distillation, resulting in more view-consistent editing. Additionally, our IESD further uses the proposed edit scale to merge two predicted scores, leading to better identity preservation and more effective editing. This approach allows our method to overcome the limitations faced by the alternative solutions, producing high-quality 3D avatars with improved fine-grained editing results.

**Limitations and failure cases.** While setting a new state-of-the-art, we acknowledge HeadSculpt has limitations, as the failure cases in Fig. 9 demonstrate: **(1)** non-deformable results hinder further extensions and applications in audio or video-driven problems; **(2)** generated textures are highly saturated and less realistic, especially for characters with highly detailed appearances (*e.g.*, Freddy Krueger); **(3)** some inherited biases from Stable Diffusion [69] still remain, such as inaccurate and stereotypical appearances of Asian characters (*e.g.*, Sun Wukong and Japanese Geisha); and **(4)** limitations inherited from InstructPix2Pix [5], such as the inability to perform large spatial manipulations (*e.g.*, remove his nose).

## 5   Conclusions

We have introduced HeadSculpt, a novel pipeline for generating high-resolution 3D human avatars and performing identity-aware editing tasks through text. We proposed to utilize a prior-driven score distillation that combines a landmark-based ControlNet and view-dependent textual inversion to address the Janus problem. We also introduced identity-aware editing score distillation that preserves both the original identity information and the editing instruction. Extensive evaluations demonstrated that our HeadSculpt produces high-fidelity results under various scenarios, outperforming state-of-the-art methods significantly.

**Societal impact.** The advancements in geometry and texture generation for human head avatars could be deployed in many AR/VR use cases but also raises concerns about their potential malicious use. We encourage responsible research and application, fostering open and transparent practices.

**Acknowledgment.** This work is partially supported by Hong Kong Research Grant Council - Early Career Scheme (Grant No. 27208022) and HKU Seed Fund for Basic Research. We also thank the anonymous reviewers for their constructive suggestions.

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

Table 2: **Hyper-parameters of HeadSculpt.**

| | | |
|---|---|---|
| **Camera setting** | $\theta$ range | (20, 110) |
| | Radius range | (1.0, 1.5) |
| | FoV range | (30, 50) |
| **Render setting** | Resolution for coarse | (64, 64) |
| | Resolution for fine | (512, 512) |
| | Max num steps sampled per ray | 1024 |
| | Iter interval to update extra status | 16 |
| **Diffusion setting** | Guidance scale | 100 |
| | $t$ range | (0.02, 0.98) |
| | $\omega(t)$ | $\sqrt{\alpha_t}(1 - \alpha_t)$ |
| **Training setting** | #Iterations for coarse | $70k$ |
| | #Iterations for fine | $50k$ |
| | Batch size | 4 |
| | LR of grid encoder | 1e-3 |
| | LR of NeRF MLP | 1e-3 |
| | LR of $s_i$ and $\Delta\mathbf{v}_i$ in DMTET | 1e-2 |
| | LR scheduler | constant |
| | Warmup iterations | $20k$ |
| | Optimizer | Adam (0.9, 0.99) |
| | Weight decay | 0 |
| | Precision | fp16 |
| **Hardware** | GPU | $1 \times$ Tesla V100 (32GB) |
| | Training duration | 1h (coarse) + 1h (fine) |

# A   Implementation details

## A.1   Details about 3D scene models

In the coarse stage, we make use of the grid frequency encoder $\gamma(\cdot)$ from the publicly available Stable DreamFusion [73]. This encoder maps the input $\mathbf{x} \in \mathbb{R}^3$ to a higher-frequency dimension, yielding $\gamma(\mathbf{x}) \in \mathbb{R}^{32}$. The MLP within our NeRF model consists of three layers with dimensions [32, 64, 64, 3+1+3]. Here, the output channels '3', '1', and '3' represent the predicted normals, density value, and RGB colors, respectively. In the fine stage, we directly optimize the signed distance value $s_i \in \mathbb{R}$, along with a position offset $\Delta\mathbf{v}_i \in \mathbb{R}^3$ for each vertex $\mathbf{v}_i$. We found that fitting $s_i$ and $\mathbf{v}_i$ into MLP, as done by Fantasia3D [74], often leads to diverged training.

To ensure easy reproducibility, we have included all the hyperparameters used in our experiments in Tab 2. The other hyper-parameters are set to be the default of Stable-DreamFusion [73].

## A.2   Details about textual inversion

In the main paper, we discussed the collection of a tiny dataset consisting of 34 images depicting the back view of heads. This dataset was used to train a special token, <back-view>, to address the ambiguity associated with the back view of landmarks. The images in the dataset were selected to encompass a diverse range of gender, color, age, and other characteristics. A few samples from the dataset are shown in Fig. 11. While our simple selection strategy has proven effective in our specific case, we believe that a more refined collection process could further enhance the controllability of the learned <back-view> token. We use the default training recipe provided by HuggingFace Diffusers [2], which took us 1 hour on a single Tesla V100 GPU.

---

[2]https://github.com/huggingface/diffusers/blob/main/examples/textual_inversion

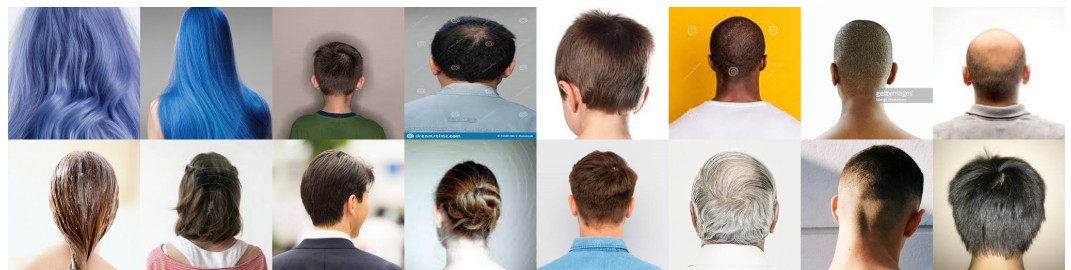

Figure 11: **Samples of the tiny dataset collected for learning** `<back-view>` **token.**

# B  Further analysis

## B.1  Effectiveness of textual inversion on 2D generation

To show the effectiveness of the learned `<back-view>` token, we conduct an analysis of its control capabilities in the context of 2D generation results. Specifically, we compare two generation results using Stable Diffusion [69], with both experiments sharing the same random seed. One experiment has the plain text prompt appended with the plain phrase "back view," while the other experiment utilizes the learned special token `<back-view>` in the prompt. We present a selection of randomly generated results in Fig. 12. The observations indicate that the `<back-view>` token effectively influences the pose of the generated heads towards the back, resulting in a distinct appearance. Remarkably, the `<back-view>` token demonstrates a notable generalization ability, as evidenced by the Batman case, despite not having been trained specifically on back views of Batman in the textual inversion process.

## B.2  Inherent bias in 2D diffusion models

In our main paper, we discussed the motivation behind our proposed identity-aware editing score distillation (IESD), which can be attributed to two key factors. Firstly, the limitations of prompt-based editing [54, 39] are due to the inherent bias present in Stable Diffusion (SD). Secondly, while InstructPix2Pix (IP2P) [5] offers a solution by employing instruction-based editing to mitigate bias, it often results in identity loss. To further illustrate this phenomenon, we showcase the biased 2D outputs of SD and ControlNet-based IP2P in Fig. 13. Modified descriptions and instructions are utilized in these respective methods to facilitate the editing process and achieve the desired results. The results provide clear evidence of the following: (1) SD generates biased outcomes, with a tendency to underweight the "older" aspect and overweight the "skull" aspect in the modified description; (2) IP2P demonstrates the ability to edit the image successfully, but it faces challenges in preserving the identity of the avatar.

The aforementioned inherent biases are amplified in the domain of 3D generation (refer to Fig. 10 in the main paper) due to the optimization process guided by SDS loss, which tends to prioritize view consistency at the expense of sacrificing prominent features. To address this issue, our proposed IESD approach combines two types of scores: one for editing and the other for identity preservation. This allows us to strike a balance between preserving the initial appearance and achieving the desired editing outcome.

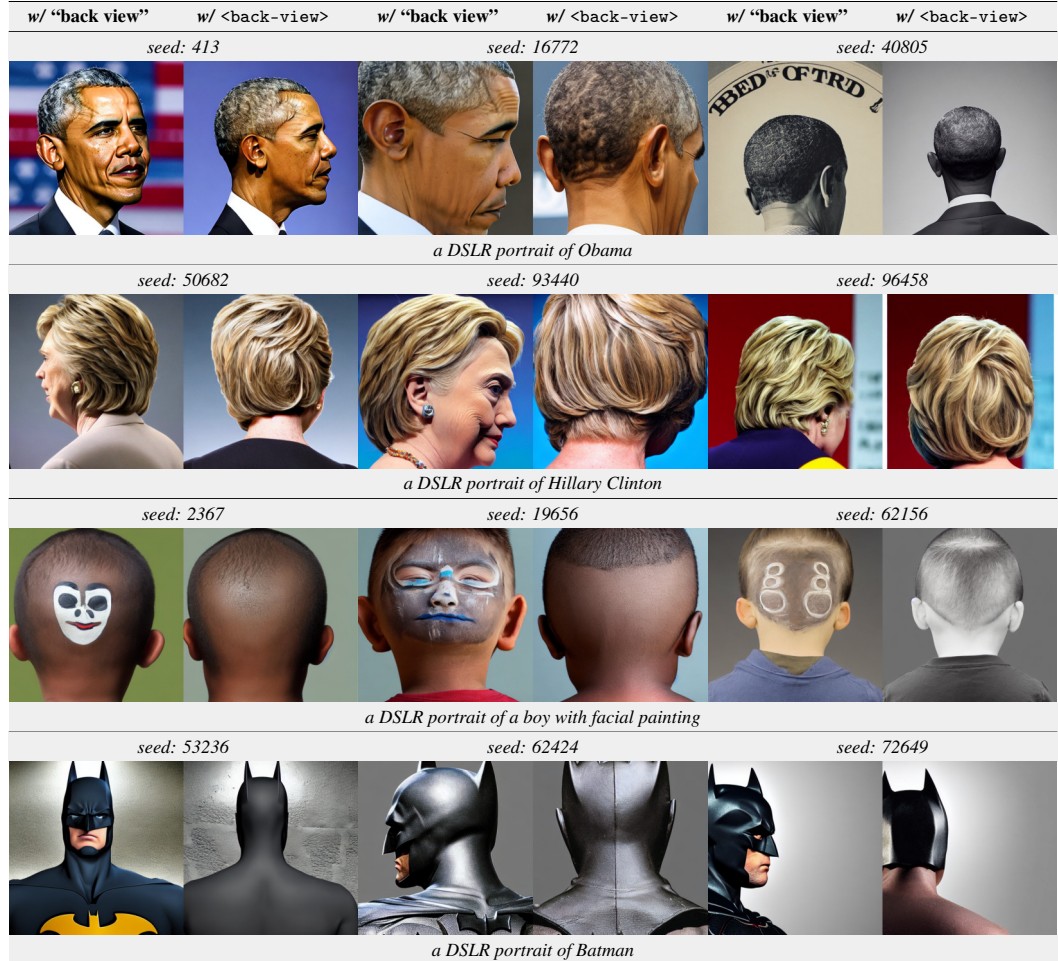

Figure 12: **Analysis of the learned `<back-view>` on 2D image generation.** For each pair of images, we present two 2D images generated with the same random seed, where the left image is conditioned on the plain text "back view" and the right image is conditioned on the `<back-view>` token.

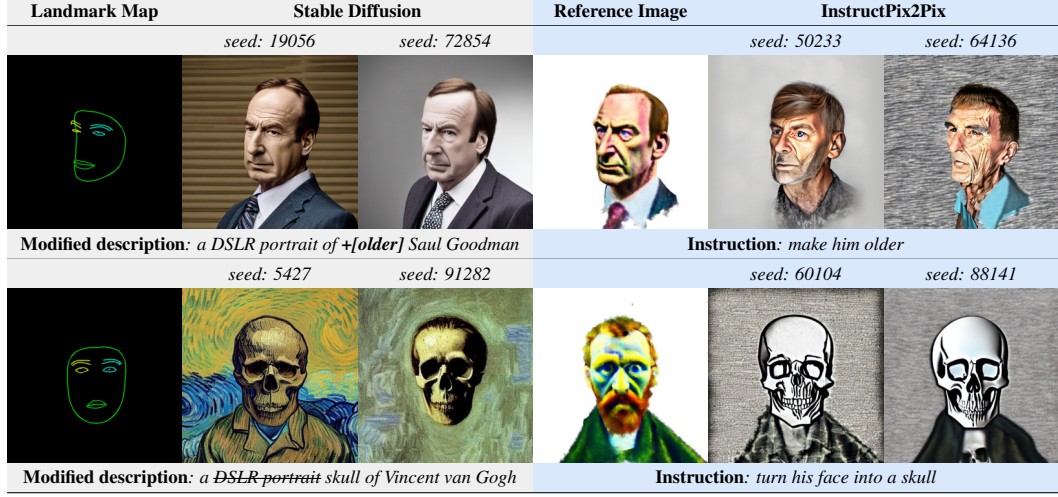

Figure 13: **Analysis of the inherent bias in 2D diffusion models.** For each case, we display several 2D outputs of SD and IP2P, utilizing modified descriptions and instructions, respectively, with reference images from our coarse-stage NeRF model to facilitate the editing process.

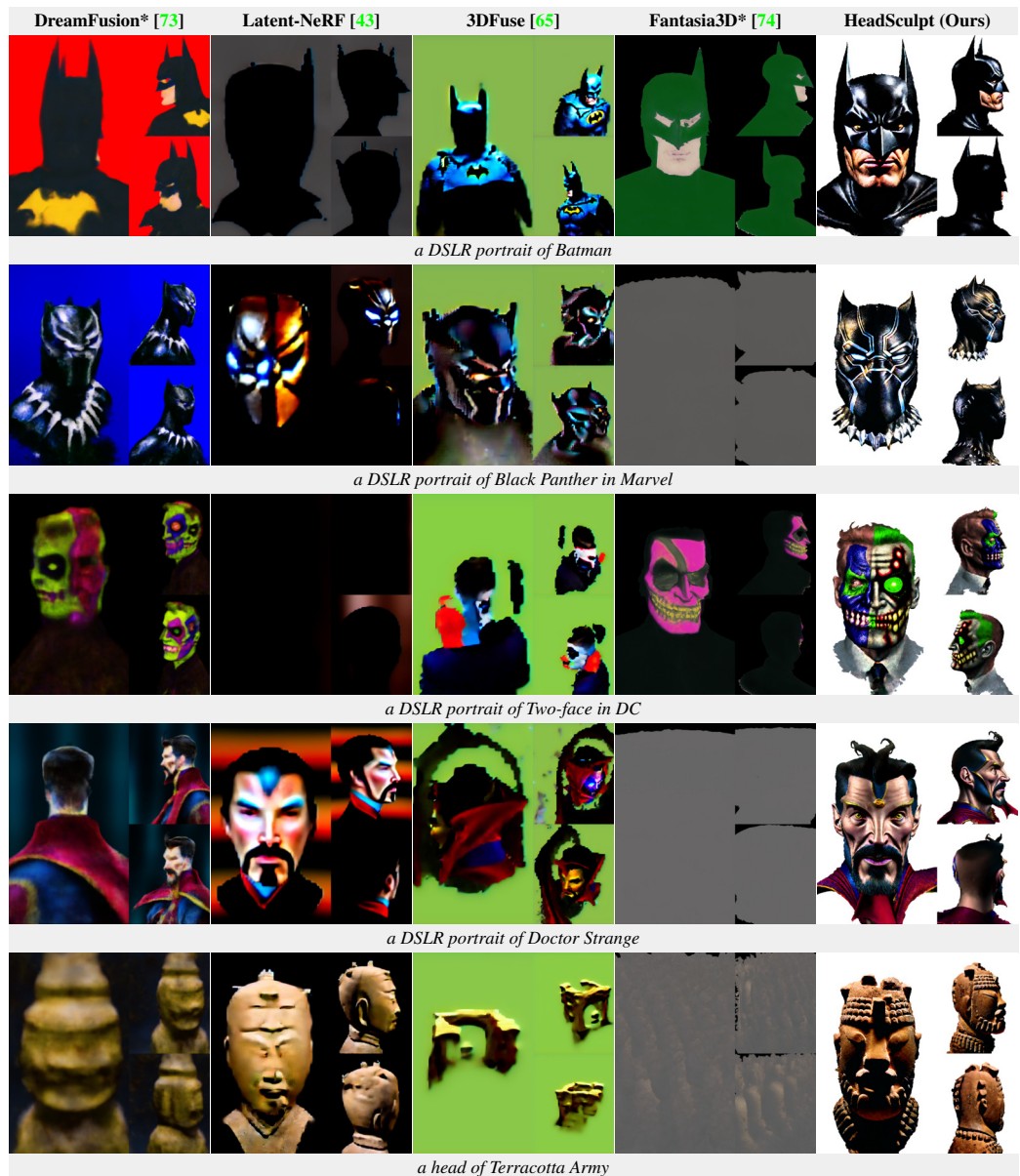

Figure 14: **Additional comparisons with existing methods on generation (Part 1).** *Non-official.

## C  Additional qualitative comparisons

### C.1  Comparison with existing methods on generation results

We provide more qualitative comparisons with four baseline methods [73, 43, 65, 74] in Fig. 14 and Fig. 15. These results serve to reinforce the claims made in Sec. 4.1 of the main paper, providing further evidence of the superior performance of our HeadSculpt in generating high-fidelity head avatars. These results showcase the ability of our method to capture intricate details, realistic textures, and overall visual quality, solidifying its position as a state-of-the-art solution in this task.

Notably, to provide a more immersive and comprehensive understanding of our results, we include multiple outcomes of our HeadSculpt in the form of 360° **rotating videos**. These videos can be accessed on `https://brandonhan.uk/HeadSculpt`, enabling viewers to observe the generated avatars from various angles and perspectives.

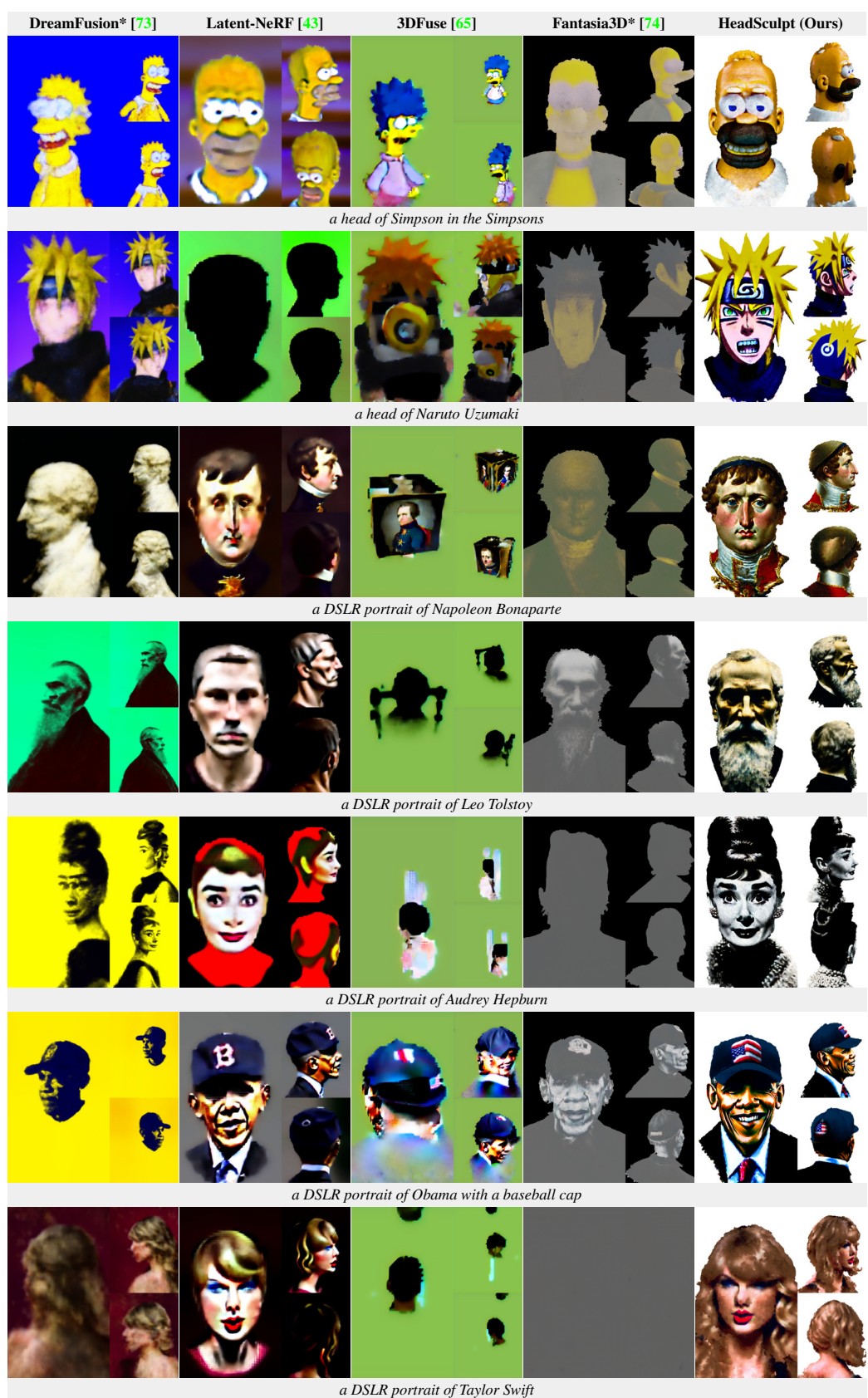

Figure 15: **Additional comparisons with existing methods on generation (Part 2).** *Non-official.

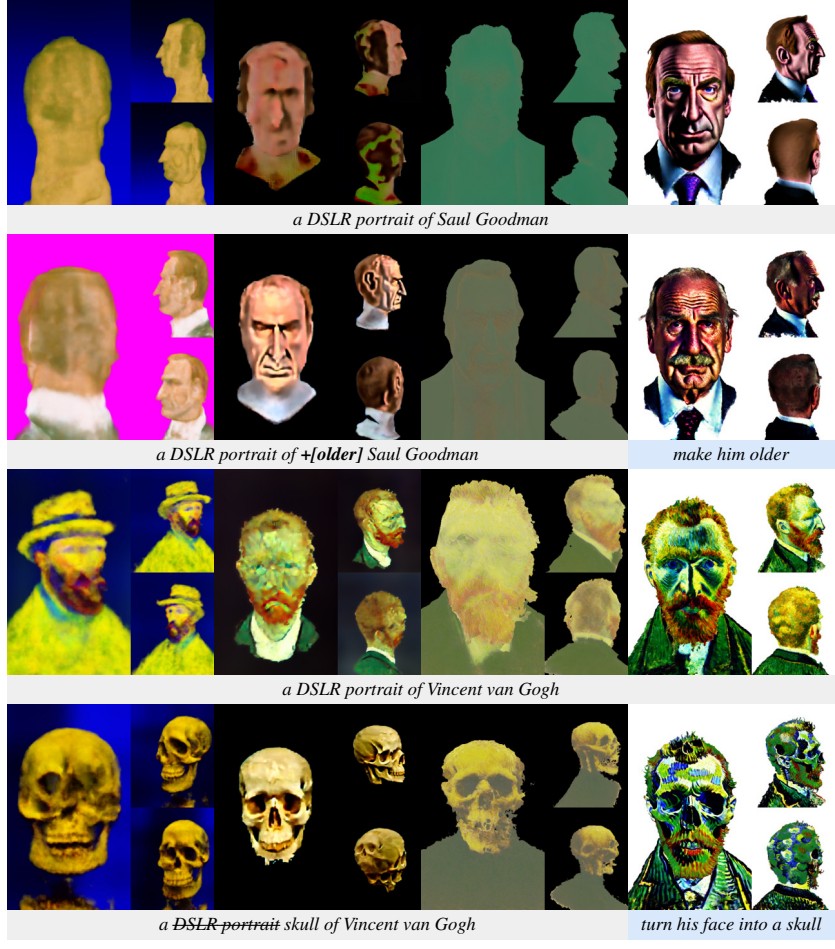

| DreamFusion* [73] | Latent-NeRF [43] | Fantasia3D* [74] | HeadSculpt (Ours) |
| --- | --- | --- | --- |

*a DSLR portrait of Saul Goodman*

*a DSLR portrait of +[older] Saul Goodman*    *make him older*

*a DSLR portrait of Vincent van Gogh*

*a ~~DSLR portrait~~ skull of Vincent van Gogh*    *turn his face into a skull*

Figure 16: **Comparisons with existing methods on editing.***Non-official.

## C.2  Comparison with existing methods on editing results

Since the absence of alternative methods specifically designed for editing, we conduct additional evaluations of the editing results generated by existing methods by modifying the text prompts. Fig. 16 illustrates that bias in editing is a pervasive issue encountered by all the baselines. This bias stems from the shared SDS guidance function, which is based on a diffusion prior, despite the variations in representation and optimization methods employed by these approaches. Instead, IESD enables the guidance function to incorporate information from two complementary sources: (1) the original image gradient, which preserves identity, and (2) the editing gradient, which captures desired modifications. By considering both terms, our approach grants more explicit and direct control over the editing process compared to the conventional guidance derived solely from the input.

| Latent-NeRF [43] | Fantasia3D* [74] | HeadSculpt (Ours) |
|---|---|---|

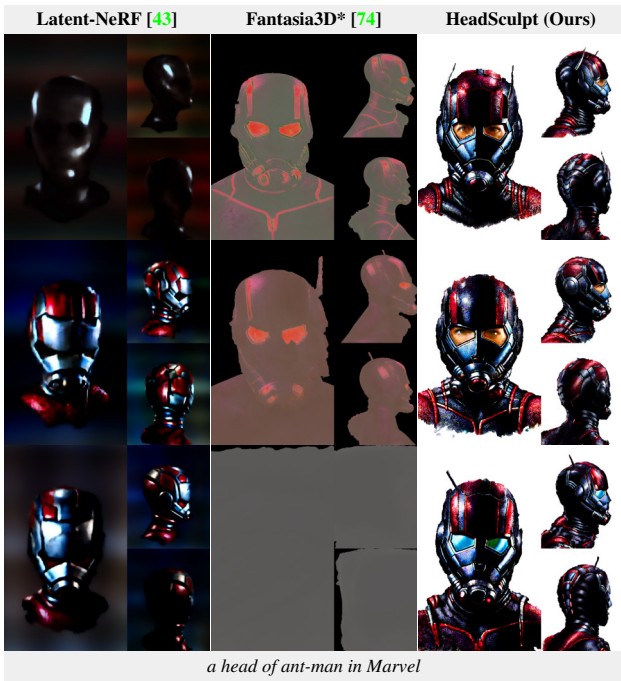

*a head of ant-man in Marvel*

Figure 17: **Results across random seeds (0, 1, 2).**\*Non-official.

## C.3   Comparison with existing methods on stability

We observe that all baselines tend to have diverged training processes as they do not integrate 3D prior to the diffusion model. Taking two shape-guided prior methods (*i.e.*, Latent-NeRF [43] and Fantasia3D [10]) as examples, we compare their generation results and ours across different random seeds. We conduct comparisons under the same default hyper-parameters and present the results in Fig. 17. We notice that prior methods need to try several runs to get the best generation while ours can achieve consistent results among different runs. Our method is thus featured with stable training, without the need for cherry-picking over many runs.

