# OpenReview forum: "HeadSculpt: Crafting 3D Head Avatars with Text"
_NeurIPS.cc/2023/Conference — NeurIPS 2023 poster_

### Official Review · Reviewer_Gcqm · 2023-06-26

**Soundness:** 3 good
**Presentation:** 3 good
**Contribution:** 2 fair
**Rating:** 6
**Confidence:** 5

**Summary:**

This work proposes a text-to-avatar creation pipeline, building upon dreamfusion and magic3d. To alleviate the geometric ambiguity, the authors replace the vanilla stable diffusion with a landmark-conditioned diffusion model finetuned with controlnet. They also use textual inversion to obtain specific token for back view. Extensive experiments show that the proposed method achieves SOTA performance.

**Strengths:**

- The generation results are impressive and compelling. As shown in Fig. 1, the method can generate avatars in different domains including photo-realistic, anime.
- The writing is easy to follow.

**Weaknesses:**

- The generated avatars lack details and suffer some artifacts. The results in Fig. 1 suffer obvious noises especially when zoomed in. Hair and clothing lack details.
- How about the generation diversity? Given the same text prompt, can the model generate multiple avatars that align with the text?
- One key aspect of avatars is animation. How could the generated avatars be driven?
- Fig. 2 is a bit confusing for me. It would be better to clean it.
- As in Fig. 3, the editing performs well for frontal view, but fails on side and back views.
- Are results in the main paper cherry-picked? Could you show some randomly sampled results? and more visualization for the geometry?

**Questions:**

- The method used DMTet in the second stage, which is a mesh-based representation. But the mesh has difficulties in representing hair, beards, and so on. How do you solve these parts?
- How do you train landmark-conditioned controlnet? training data?

**Limitations:**

Authors have claimed the limitations.

---

> ### Author Rebuttal · Authors · 2023-08-09
>
> # 5. Response to reviewer `#Gcqm`
> We thank reviewer `#Gcqm` for agreeing with our motivation and acknowledging the results as "impressive and compelling".
> Below we respond to the doubts put forward by the reviewer - **we regularly refer to the general response above and the provided one-page pdf with figures:**
>
> ## 5.1. Noises and artifacts in generated avatars
>
> Please refer to Sec. 1.3 of the above general response.
>
> ## 5.2. Generation diversity
>
> Indeed, like existing text-to-3d methods (e.g., DreamFusion), our method also does not yield large amounts of diversity across random seeds (as shown in Fig. 11). This is likely due to the mode-seeking property of $\mathcal{L}_{\mathrm{SDS}}$ combined with that at high noise levels, the
> smoothed densities may not have many distinct modes. Understanding the interplay between guidance strength, diversity, and loss functions remains an important open direction for future research.
>
> ## 5.3. Avatar animation
>
> Please refer to Sec. 1.1 of the above general response.
>
> ## 5.4. Cleaning Fig. 2
> Thanks for pointing this out. We will reorganize the layout and clean it in the revised version.
>
> ## 5.5. Editing performs badly on side and back views
>
> This is because the front view of a head represents almost the whole information for editing, the editing results of side and back views sometimes are inferior to those of front views. However, we respectfully argue that this is not always the case, e.g., most of the editing results as shown in Fig. 1 are similarly satisfactory among all views.
>
> ## 5.6. Randomly sampled results
>
> > Are results in the main paper cherry-picked? Could you show some randomly sampled results?
>
> The results shown in the submitted manuscript are not cherry-picked. Please also check the newly added results across random seeds in Fig. 11, where we don't find obvious geometry and appearance differences among different runs.
>
> > and more visualization for the geometry?
>
> For geometry visualization, we have now provided additional normal-rendered images in Fig. 12. Please find more examples in the supplementary file.
>
> ## 5.7. Difficulties of mesh in representing hair and beards
>
> In this work, we didn't design specific modules to handle the issue of hair and beard (out of our current focus and scope). We also note that, compared with pure NeRF-based methods, using mesh as a second-stage representation can offer sharper and more structural hair and beard, since the initialization from NeRF's density can be further trained to improve at a much higher resolution. To further improve the quality of hair and beard, one possible solution for future work is building high-quality hair/beard templates, which can be disentangled from the whole optimization process.
>
> ## 5.8. Details about landmark-conditioned ControlNet
>
> As mentioned in L167, instead of training ControlNet by ourselves, we took an off-the-shelf version[1] trained on LAION-Face[2] dataset, including 50M diverse face images and the corresponding face landmarks predicted by Google MediaPipe.
>
> [1] ControlNetMediaPipeFace. https://huggingface.co/CrucibleAI/ControlNetMediaPipeFace.
>
> [2] General facial representation learning in a visual-linguistic manner.

---

> ### Comment · Reviewer_Gcqm · 2023-08-18
>
> Thanks to the authors for their responses. After reading the rebuttal and comments from other reviewers, I would keep my original ratings.

---

### Official Review · Reviewer_thuE · 2023-07-03

**Soundness:** 3 good
**Presentation:** 3 good
**Contribution:** 2 fair
**Rating:** 6
**Confidence:** 5

**Summary:**

Existing text-driven 3D generative models could have many problems, such as geometric artifacts (e.g., Janus problem),  and visual inconsistency.  This work focuses on 3D head avatar generation, which utilizes FLAME model to incorporate human geometric priors into generation, hence resolving those problems in 3D generation.  The proposed framework utilizes a coarse-to-fine framework, and it first generates a coarse face via the neural radiance field (NeRF) and then performs refinement/editing using tetrahedron mesh (DMTet).  As a result, the proposed method could generate diverse 3D human faces, and it also enables identify-aware editing with the help of both ControlNet and InstructPix2Pix.

**Strengths:**

(1) This work proposes a text-guided 3D generation, specifically for human heads.  It could successfully generate various human heads by using or modifying many existing methods, such as FLAME, ControlNet, Instruct-Pix2Pix, and DMTet.

(2) This method also studies identity-aware editing by defining a trade-off between the original appearance and the desired editing.

**Weaknesses:**

The proposed text-guided 3D generation method focuses on human heads, but it seems like combining 3D head priors with the existing general 3D generation or style-editing methods.  A few problems associated with 3D human heads have not been deeply studied, and please refer to the following detailed weaknesses:

(1) This method effectively addresses the limitations encountered in current text-guided 3D generation approaches through the integration of 3D head priors with the FLAME model. By leveraging the FLAME model for 3D head generation, it is expected that this method could facilitate intricate expression variations.  However, the current generated head avatars seem to be static only, and the manuscript did not provide many expressions except for the neutral expression.  Enabling expressions is important for 3D head avatars, and a few examples could be found in the FLAME model, ICT-FaceKit, and the very relevant work DreamFace.

(2) The identity-aware editing has not been well-justified.  Based on my understanding, the current method simply mixes the style of the original appearance and the desired editing to generate the facial texture based on the ControlNet-based InstructPix2Pix.  I am wondering if it is possible to do a few specific editing of human heads, such as changing the hairstyle, wearing different glasses, and perhaps a few other geometric adjustments (e.g., getting fatter or getting thinner).

**Questions:**

Besides the questions listed in the weakness, I would like to ask:
(1) Is the generated head avatar capable of performing various expressions?  Or is it possible to perform different expressions by using simple post processes?

(2) Is there a clear definition of the so-called identity-awareness of the editing process? Merely blending styles appears to be an unsatisfactory solution.

(3) Is it possible to generate/edit human head avatars with different shapes?

**Limitations:**

Yes.

---

> ### Author Rebuttal · Authors · 2023-08-09
>
> # 4. Response to reviewer `#thuE`
> We are grateful the reviewer `#thuE` took the time to thoroughly review our work and found that our method "effectively addresses the limitations encountered in current text-guided 3D generation approaches".
> We provide our responses regarding the raised concerns as follows  - **we regularly refer to the general response above and the provided one-page pdf with figures:**
>
> ## 4.1. Expression variations
>
> > the current generated head avatars seem to be static only, and the manuscript did not provide many expressions except for the neutral
>
> We acknowledge our current method does not enable direct animation, as discussed in the limitations. This is the sacrifice of increasing generalization ability, where we didn't optimize in the pre-trained parametric space of FLAME while only using it as the density initialization of NeRF. However, expression edits are possible using IESD, as Fig. 9 shows. Despite lacking direct animation, our edited expressions demonstrate a reasonably consistent appearance and identity.
> Please refer to Sec. 1.1 of the above general response for more details.
>
> ## 4.2. Specific editing
>
> > I am wondering if it is possible to do a few specific editing of human heads, such as changing the hairstyle, wearing different glasses
>
> In Fig. 10, we have now provided several additional editing results as suggested, including hairstyle change, beard change, and adding sunglass. It is evident that our method can achieve satisfactory editing results.
>
> >  and perhaps a few other geometric adjustments (e.g., getting fatter or getting thinner)
>
> > Is it possible to generate/edit human head avatars with different shapes?
>
> We found it is generally difficult to do geometric adjustments via editing since it is challenging and ambiguous to describe a desired geometry via text prompts alone. However, thanks to our NeRF initialization from the FLAME model and corresponding landmark control in the diffusion prior, our method supports geometric variation in generation by changing the FLAME model. We only used the canonical FLAME model by default in the submitted manuscript. To demonstrate our method's ability for geometric adjustment, we have further provided several examples with different FLAME models as the initialization, as shown in Fig. 12. We will add this result in the revision.
>
> ## 4.3. Definition of IESD
>
> > Is there a clear definition of the so-called identity awareness of the editing process?
>
> We define identity awareness as the ability to make desired modifications to an avatar's appearance based on editing instructions while preserving key facial features and attributes that represent the avatar's core identity and preventing undesired changes unrelated to the edit.
> As introduced in L202-213, the proposed IESD is a variant of SDS, characterized by blending two different scores for noise prediction.
>
> > Based on my understanding, the current method simply mixes the style of the original appearance and the desired editing to generate the facial texture based on the ControlNet-based InstructPix2Pix.
>
> We note that it is more than a style mix of the original appearance and the desired editing, because the gradient predicted by InstructPix2Pix for editing is dominant in the desired editing area, so it's capable of local region editing, as shown in Fig. 10.

---

> > ### Comment · Reviewer_thuE · 2023-08-19
> > **Response to the Rebuttal and additional questions**
> >
> > Thanks for the rebuttal.  After reading the other reviewers' comments, I realize we have similar concerns about the method pipeline.
> > As for the method details, I have more questions, and I hope the authors can provide a thorough explanation.
> >
> > (1) How is the "Mixing" in Figure 2 implemented? The manuscript does not seem to provide corresponding formulas or descriptions for it.
> >
> > (2) In P5 L169,  what are the criteria for selecting vertices? Is it random or algorithmic?
> >
> > (3) In P6 L186, where was the data collected from? Is it public or private? Was the selection criteria completely random?
> >
> > (4) In P6 L207, do I and C correspond to the two ControlNets in Figure 2(c) of the PSD? The authors claim they are identical, but based on what is mentioned in section 3.3 and the original ControlNets, they should represent different tasks.
> >
> > (5) In P6 L211, according to Figure 2, the reference image should be renderable from both DMTET (high-resolution) and Coarse Nerf. Why is it rendered only from Coarse Nerf in this case?
> >
> > I will raise my rating if all unclear aspects are adequately explained.

---

> > > ### Author Response · Authors · 2023-08-19
> > >
> > > Thanks for your reply. We provide explanations with respect to these additional questions as follows:
> > >
> > > 1. Apologies for the confusion. The "Mixing" in Figure 2 refers to Equation 7, which is the weighted averaging of the two predicted noises for IESD.
> > > 2. These vertices are selected by a pre-defined index set, which includes the vertices that correspond to the contour of face, eyes, nose, and mouth. This selection process is used to process the projected dense landmarks in the same style as the sparse landmark maps used for training ControlNet. This procedure is the same in all experiments.
> > > 3. The images in the tiny dataset are manually downloaded from Google search. Please find more details and example images in the submitted supplementary material.
> > > 4. Yes, I and C correspond to the two ControlNets in Figure 2(c) of the PSD. We apologize for any misunderstanding - we did not claim I and C are identical. We would be grateful if reviewer `#thuE` could elaborate so we can address this point fully.
> > > 5. As shown in Figure 2, we use a frozen coarse NeRF to render reference images and a fine DMTET to learn the newly edited 3D representation. This means that frozen coarse NeRF-rendered images do not change, while fine DMTET-rendered images do. If we use the DMTET-rendered images as the reference images, the reference image at $i$ iteration will be the edited result from $i-1$ iteration, which will endlessly accumulate the editing appearance and thus leads to unsatisfactory results.
> > >
> > > Please let us know if these explanations address your questions. We remain open to further discussions and warmly welcome any additional feedback or inquiries you might have.

---

> > > > ### Comment · Reviewer_thuE · 2023-08-21
> > > >
> > > > Thanks for the answers, and I have raised my rating.

---

### Official Review · Reviewer_ztub · 2023-07-03

**Soundness:** 3 good
**Presentation:** 4 excellent
**Contribution:** 3 good
**Rating:** 6
**Confidence:** 4

**Summary:**

The paper proposes a novel method for performing text-to-3D generation for specifically human (or humanoid) heads. These generated heads can be further edited and refined using more fine-grained detailed text prompts while still preserving the identity of the generated asset. This is accomplished with two main contributions: using a landmark-aware score distillation loss in order to ensure that the generated heads roughly align with a template head mesh (FLAME), and introducing a loss function which balances text-based editing with a new description with the original description used to generate the asset. The generated results fix a number of artifacts with existing text-to-3D generation methods specifically for heads. For example, the generated heads do not suffer from the multi-face problems, and the strong prior given by a parametric head model ensures that the generated results are geometrically correct. Additionally, the results can be edited while still maintaining the identity of the original generated results. This is shown in comparison to a number of baseline methods, which are outperformed.

AFTER REBUTTAL:

I have read the rebuttal and appreciate the detailed response. I appreciate the additional quantitative evaluations, as I believe this was a limitation of the method that has been addressed. However, I still think the quality is a bit limited, but this is the case for most text-to-3D methods. The additional comparisons weakness has been addressed as well. However, I don't see justification for the the more stable training - I would suggest removing this claim without verifying it significantly, while Fig 11 is nice, I am not sure this is enough to make an entire claim based on.

**Strengths:**

In my opinion, the strengths of the paper are that:
1. The presentation of the paper is extremely high quality. The introduction and related work sections are comprehensive, motivate the problem well, and clearly delineate the contributions provided in the paper. The methods section is very clearly described and is simple to follow for those familiar with the field of generative 3D. I view this as very important because it is much easier to glean information from the paper, and future work in text-to-3D (potentially for avatar generation) would be much more likely to build off of this method.
2. The contributions are clearly stated, and to the best of my knowledge they are novel, and they seem like they could be relatively significant.
    - The idea of integrating ControlNet together with a score distillation loss makes intuitive sense for gaining better control over generated 3D assets, and I have not, to the best of my knowledge, seen it proposed elsewhere. Maintaining control over the structures which are generated from text-to-3D, including ensuring they have higher quality and less noisy geometry, is a very important problem in text-to-3D generation and I view this solution as potentially having a large impact on those working in this field.
    - While balancing losses between an editing prompt and original prompt seems simple, it does not seem to be used in other text-to-3D methods and seems like it could be useful for text-based editing of already fitted representations in other applications.
3. I find the ablation study to be high quality, and ablate all of the parts of the method which are relevant: introducing ControlNet into the SDS loss, the representation chosen and coarse-to-fine optimization, and the hyperparameters for balancing the editing losses. These are all clearly ablated, along with other smaller contributions such as textual inversion for generating the back of the head.

**Weaknesses:**

In my opinion, there is only one main weakness with the paper, which is that the evaluations and comparisons to baseline methods do not seem complete.
1. There are no quantitative comparisons. While the user study is insightful, methods like DreamFusion proposed the CLIP-R metric. Is there a reason why this was not used in this paper? Does the metric do a poor job at quantifying what it is trying to measure? Additionally, some quantitative results for evaluating the editing quality would be useful: CLIP-R (or something like it, if CLIP-R is not useful) can be used to ensure that the identity is still preserved despite editing with an additional prompt (and perhaps also show that the rendered images are pushed closer to a desired fine-grained editing prompt).
2. The baseline methods are not evaluated entirely fairly. For example, I don’t see any of the baseline methods evaluated for editing quality. What happens if one of the generated examples (ex: those from Fig 4) go through a fine-grained text-based edit? How much will the identity be degraded or the edit not be done sufficiently? I think this is extremely important to show because without it, I don’t know what is the state-of-the-art that is currently being improved on and thus don’t know if the proposed method is even better.
3. It is mentioned that the optimization of the proposed method is “more stable” (L244-245) than baseline methods. However, this claim is not justified anywhere. This would be an extremely important contribution for text-to-3D (where optimization is notoriously brittle), so some quantification of this would be very impactful if it could be shown. For example, ranges of hyperparameters trained with which still lead to a good solution, or error bars on generated results, or failure cases.

**Questions:**

I do not have additional questions brought up (see weaknesses for evaluations questions). The paper is exceptionally clear in describing the method and contributions.

**Limitations:**

The paper adequately addresses the limitations of the proposed method. The first limitation, which mentions that the results are “non-deformable” is insightful as this was a question which is natural to ask considering that the FLAME template is a deformable model for heads. Additionally, the generated results still are not completely photorealistic, such as those representations which have been fit from captured data. Some additional failure cases of the method would be interesting to see in order to understand better what the limitations are of the method and which piece would be the biggest bottleneck for someone who wanted to integrate this work into their application.

---

> ### Author Rebuttal · Authors · 2023-08-09
>
> # 3. Response to reviewer `#ztub`
> We really appreciate the feedback provided by reviewer `#ztub`.
> Thanks for finding our presentation clear and our contributions novel and regarding our method as a fundamental basis that future work could "build upon".
> Below we address the proposed concerns about  evaluations and comparisons  - **we regularly refer to the general response above and the provided one-page pdf with figures:**
>
> ## 3.1. More quantitative evaluation
> Please refer to Sec. 1.4 of the above general response.
>
> ## 3.2. Baseline methods evaluated for editing quality
> As suggested, we have now provided several editing results produced by the baselines via prompt modification. As shown in Fig. 13, bias in editing is a common problem that all baselines suffer. This is because although variation in the representation and optimization methods, these methods share the same guidance function (i.e., $\mathcal{L}_{\mathrm{SDS}}$) of diffusion prior, where the bias comes from. In this paper, we propose IESD, which allows the guidance function to incorporate information from two complementary sources: 1) the original image gradient that preserves identity and 2) the editing gradient that captures desired modifications. By factoring both terms, our IESD enables more explicit and direct control over the editing process compared to the conventional guidance derived from the input alone.
> We will further clarify this in the revised version.
>
> ## 3.3. Stability comparisons
>
> > It is mentioned that the optimization of the proposed method is “more stable” (L244-245) than baseline methods. However, this claim is not justified anywhere.
>
> We observed that all baselines tend to have diverged training processes as they do not integrate 3D prior to the diffusion model. Taking two shape-guided prior methods (i.e., Latent-NeRF and Fantasia3D) as examples, we compare their generation results and ours across different random seeds. We conduct comparisons under the same default hyper-parameters and present the results in Fig. 11. We can observe that prior methods need to try several runs to get the best generation while ours can achieve consistent results among different runs. Our method is thus featured with stable training, without the need for cherry-picking over many runs.
>
> > For example, ranges of hyperparameters trained with which still lead to a good solution
>
> All the experiments, for all different methods, presented in the submitted manuscript and rebuttal were conducted under the same default hyper-parameters.
>
> ## 3.4. Failure cases and bottlenecks
>
> Thanks. We have now given several failure cases in Fig. 14. As discussed in the limitation part of the manuscript, our method is not perfect, for example, 1) it inherits the bias from the diffusion model, e.g., it cannot correctly generate characters in the eastern culture like "Sun Wukong"; 2) it can not handle highly-detailed textures due to the mode-seeking property of $\mathcal{L}_{\mathrm{SDS}}$, e.g., "Freddy Krueger".
>
> > which mentions that the results are “non-deformable” is insightful as this was a question which is natural to ask considering that the FLAME template is a deformable model for heads.
>
> Please refer to Sec. 1.1 of the above general response.
>
> > the generated results still are not completely photorealistic
>
> Please refer to Sec. 1.2 of the above general response.

---

> > ### Comment · Reviewer_ztub · 2023-08-17
> >
> > Thank you for the detailed clarification and additional quantitative and qualitative evaluations of the method and baselines. This has certainly improved my opinion on the evaluation quality of the method. I do not have additional questions on this.
> >
> > Overall, I am now convinced the quality is an improvement over state-of-the-art, at the cost of limiting to a specific class of objects: heads. However, I do feel that the lack of being able to animate the result severely limits the applicability: a method specifically designed for heads should be able to include head-specific features such as expression editing. With both of these in mind, I still feel slightly positively about the paper in its current state.

---

> > > ### Author Response · Authors · 2023-08-18
> > >
> > > Thank you for your reply - we are pleased our responses addressed your questions.
> > >
> > > We agree with you that animation is valuable for downstream tasks, but it should not compromise performance or generalization capabilities. At this stage, our framework supports expression editing, albeit not full animation. Enabling animation while retaining generalization ability remains an important challenge as discussed above. Exploring optimal solutions to unlock animation without sacrificing quality or generalization will be a key focus in our future work.  Two potential solutions we aim to investigate are: 1) finding improved head representations that retain mesh structure to enable animation while maintaining generalization capacity; and 2) utilizing auto-animation tools in the off-the-shelf graphic pipeline as post-processing on current outputs to add motion. Your insightful comments will help strengthen our work. We are grateful for your time and input.

---

### Official Review · Reviewer_ooKH · 2023-07-05

**Soundness:** 4 excellent
**Presentation:** 4 excellent
**Contribution:** 3 good
**Rating:** 7
**Confidence:** 4

**Summary:**

Authors proposed a solution for generating human heads based on text description. The technology is based on popularised concepts of control signal for pre-trained diffusion model and static 3D model to guarantee consistent head. Moreover, authors presents the concept of the “back” of the head with a textual inversion. The qualitative and quantitative evaluation shows the superiority of the method over existing competitors.


**Strengths:**

- there are plenty of interesting ideas how to do specific solution including domain knowledge: 1) statistic 3d model (e.g. FLAME), 2) improved score distillation with additional special token, 3) editing directions
- Authors demonstrate a wide set of experiments to evaluate their method including small use study


**Weaknesses:**

There is a huge limitation of the method compared to the baselines
- The user study can be biased - small set of people, without description of the task and based on the figure 4 it is extremely easy to select the introduced method.
- Images are not photorealistic, furthermore the results have an disdvantage from the volumetric rendering (small noise is still visible on all counters)
- Results are biased in many directions: huge emotion bias despite the neutral emotions (Figure 1), colour issue for the shadows and unrealistic light baked to the texture.


**Questions:**

Most of the following questions can improve the manuscript:
- Why do you use  DMTet for discretization? The motivation is missing at line 144+
- In the paragraph near L155 you are elaborating the source of the existing issues. Do you think that main problem is an ambiguity and with rendering each representation for uniformly sample cameras will help to disambiguate the gradients during denoising?
- How does the back-view concept is so important if you introduce the landmark based controlNet and the FLAME structure? Can you elaborate why it is not enough to use just the full head with landmarks? The Figure 6 slightly contradicts original motivation of the textual inversion concept for the “back”.
- Could you explain in more details the limitation from InstructPix2Pix (L305)?


**Limitations:**

- All faces only in a neutral position that is not always correlate with the textual description (e.g. neutral clown)
- The overall image quality is biased to the unrealistic colours
- Most of the ideas will work only for the presented setup of the heads

---

> ### Author Rebuttal · Authors · 2023-08-09
>
> # 2. Response to reviewer `#ooKH`
> We thank reviewer `#ooKH` for recognizing that our solution is "interesting" and our qualitative and quantitative evaluation "shows superiority".
> We address the proposed questions as follows - **we regularly refer to the general response above and the provided one-page pdf with figures:**
>
> ## 2.1. Details about the user study
> We conducted the user studies in the form of a questionnaire supported by Google Forms. Each volunteer will be presented with 20 randomly selected generated results with rotating videos, with the description to be:
>
> ```You will be presented with 20 sets of results with each generated by 5 text-to-3D methods. Your task is to evaluate and rank the results based on three dimensions: consistency with the provided prompt, texture quality, and geometry quality. For each set of results, please assign a score of 5 to the best candidate and 1 to the worst, with intermediate scores for the others.```
>
> ### 2.1.1. Updated results with additional volunteers
>
> >The user study can be biased - small set of people
>
> To reduce the potential bias in the user study, we have now expanded the evaluation to include additionally 22 volunteers, bringing the total number of participants to 42. The updated user study statistics are presented in Fig. 8,  indicating that our method consistently achieves the highest ranks.
>
> ### 2.1.2. Additional quantitaive evaluations
>
> To present more comprehensive evaluations, we have also now conducted a quantitative evaluation using CLIP-R and CLIP-Score as the objective metrics. The results, provide further quantitative evidence that supports the subjective user study's findings. Please refer to Sec. 1.4 of the above general response for more details.
>
> ## 2.2. Noises on the textures
>
> Please refer to Sec. 1.3 of the above general response.
>
> ## 2.3. Biased results in emotion and color
>
> Please refer to Sec. 1.2 of the above general response.
>
> ## 2.4. Motivation of DMTet for discretization
>
> We opted for a mesh-based approach to achieve enhanced resolution in the fine stage. Yet, when explicit meshes are directly derived from implicit representations like NeRF using marching cubes, they may yield lower-quality surfaces, especially for the boundary. Introducing DMTet, a differentiable mesh representation, into the optimization pipeline offers the potential to refine the geometry extracted from NeRF. We will elaborate this consideration in the forthcoming revised edition.
>
> ## 2.5. Source of the existing issues
>
> The main reason for the Janus problem is the absence of 3D prior in the diffusion model, since it is trained on 2D images without camera pose conditioning.
> Better sampling camera poses might solve this problem to some extent, but a relatively large batch size is needed to sample as many poses as possible to better alleviate the ambiguity (e.g., Fantasia3D uses 24 as batch size over 8 Nvidia RTX 3090 GPUs to guarantee a satisfactory result).
> Instead, we focus on solving the core reasons for the Janus problem by proposing to integrate the 3D head prior to the pre-trained diffusion model via the projected facial landmarks. This proposed solution offers improved training efficiency and is more user-friendly.
> We will further explain this in the forthcoming revised edition.
>
> ## 2.6. Importance of $\texttt{\<back-view\>}$ concept
>
> Thanks for this question. Because the pre-trained diffusion model only supports 2D information conditions, we must project 3D landmarks to a 2D landmark map. However, the projection brings ambiguity over front and back views. Concretely, since the 2D image dataset used for training landmark ControlNet mostly contains only front or side face views, the model tends to generate a front-view face (i.e., a dataset bias), given such an ambiguous 2D landmark map. The proposed $\texttt{\<back-view\>}$ concept is designed exactly to resolve this bias.
>
> >  Can you elaborate why it is not enough to use just the full head with landmarks?
>
> Using full-head landmarks and considering their self-occlusion might be a possible solution, but this would require more robust full-head landmark registration (hard to collect), along with the need for retraining the ControlNet on extra data. Instead, as a cheaper solution, we leverage available facial landmarks and ControlNet capabilities in this work. We will explore more elegant landmark usage in future work.
>
> >The Figure 6 slightly contradicts the original motivation of the textual inversion concept for the “back”.
>
> For the comment regarding Fig. 6, we intend to show that landmark control is relatively more important than the $\texttt{\<back-view\>}$ concept.
> Without landmarks, it will be hard to distinguish different views and thus tend to generate always biased front views even with the help of $\texttt{\<back-view\>}$ concept.
> In summary, landmarks should play a more central role in conveying 3D pose information to the 2D diffusion process, whilst the $\texttt{\<back-view\>}$ concept is also indispensable for alleviating the ambiguity brought by the projection process. We will further clarify this in the revised manuscript.
>
> ## 2.7. Limitation from InstructPix2Pix
>
> As mentioned in the original paper, InstructPix2Pix is limited by the visual quality of the Prompt2Prompt-generated dataset. Specifically, it struggles with viewpoint changes, makes excessive image alterations, fails to isolate objects in cases, and cannot easily reorganize or swap objects. These limitations carry on our pipeline since we directly use InstructPix2Pix as is.
> We showcase an unsatisfactory editing example in Fig. 14. Importantly, our framework can easily incorporate any improvements made in the InstructPix2Pix and its follow-up works, e.g., MagicBrush[1].
>
> [1] MagicBrush: A Manually Annotated Dataset for Instruction-Guided Image Editing.

---

> > ### Comment · Reviewer_ooKH · 2023-08-21
> > **Reply and final note.**
> >
> > Thanks to the authors for their responses, I hope most of the common unclear moments will be added to the final text. After reading the rebuttal and comments from reviewers, I would keep my original ratings.

---

### Author Rebuttal · Authors · 2023-08-09

We deeply appreciate the reviewers' thoughtful feedback recognizing the presentation, novelty, and performance of our method.
The reviewers suggested additional experiments and illustrations to highlight strengths, clarify limitations, and illustrate future directions. We are pleased to have conducted all of these valuable recommended experiments, as outlined in the general and per-reviewer responses. Next, we would like to first provide some general responses to address common questions. **To avoid any confusion with the figures in the original submission, we have numbered the new figures in the attached document starting from Fig. 8 onwards.**

# 1. General response

## 1.1. Avatar expressions [Reviewers `#ztub`, `#thuE`, `#Gcqm`]
In its current form, our method does not support direct animation of the generated avatars, despite enabling barely satisfactory facial expression changes through editing (as shown in Fig. 9). This limitation arises because we initialize the NeRF density using only the FLAME model, without preserving the mesh structure and point correspondence. Our motivation for this design choice is to maximize the generalization capability across the entire pipeline. In principle, we could employ any head parametric model (e.g. NPHM[1]) as the 3D representation and optimize an avatar within its parametric space (i.e. shape and expression parameters). However, by constraining the optimization to a pre-trained parametric space in this manner, it would restrict the ability to generalize to out-of-distribution data (e.g. non-human-like avatars). We believe there is an inherent trade-off between generalization ability and structural consistency that warrants further investigation. As future work, we will explore solutions for the best compromise between these two competing objectives.

## 1.2. Biased results [Reviewers `#ooKH`, `#ztub`]
Indeed, the current pipeline exhibits biases in aspects like color, appearance, and emotion inherited from two sources: 1) the diffusion prior, which tends to memorize training data during generation[2]; and 2) the mode-seeking nature of the SDS loss, which requires large CFG values to boost fidelity at the cost of high saturation and unrealistic colors. As demonstrated in the manuscript, these biases plague all existing text-to-3D methods but are not specific to our model, though IESD can partially mitigate them during editing. By our knowledge and experience, some concurrent works might be able to alleviate these biases: a stronger diffusion model[4], image reconstruction loss[5], and more advanced score distillation[6].

## 1.3. Noises and artifacts in generated avatars [Reviewers `#ooKH`, `#Gcqm`]
We do acknowledge that the current results are far from perfect even though our results have already outperformed prior alternatives by a large margin. This is still an unsolved problem with a heavy need for further investigation and innovation. More specifically, this is due to the challenging mode-seeking nature of this zero-shot task, where real albedo textures are not available for model optimization. Compared with NeRF-based methods rendered at a much smaller $64\times 64$ resolution, the noises in our setting become more apparent because we are using mesh-based representation at an $8\times$ higher resolution at the fine stage. We hypothesize that further fine-tuning the diffusion prior on a large dataset with manually collected albedo maps[3] could help mitigate these issues, which we leave for future work. While not yet perfect, we believe our method demonstrates strong zero-shot performance and establishes a promising direction for high-fidelity generative avatar modeling without albedo supervision. We argue this already makes a significant contribution to the community.

## 1.4. More quantitative evaluation [Reviewers `#ooKH`, `#ztub`]

### 1.4.1 Generation evaluation
We appreciate the insightful suggestion. Following the suggestion, we computed CLIP-R (CLIP-L/14) and CLIP-Score metrics for all methods using the 30 text prompts from the generative process. As shown in Tab. 1, our approach substantially outperforms the competitors on both metrics. This aligns with and further substantiates the subjective superiority demonstrated in the user study. We will include this evaluation in the revised version.

### 1.4.2 Editing evaluation
We didn't provide a quantitative evaluation for editing because we consider editing to be fundamentally a subjective task and also there is no standard protocol to measure its performance. Regardless, as suggested, we have now adopted CLIP Directional Score as described in InstructPix2Pix and StyleGAN-NADA to measure the editing performance.
It measures how much the change in text captions agrees with the change in the images. Concretely, for the directional score, we encode a pair of images (the original and edited 3D models, rendered at a given viewpoint), as well as a pair of text prompts that describe the original and edited scenes, e.g., "a DSLR portrait of Saul Goodman" and "a  DSLR portrait of Saul Goodman dressed like a clown". We compare our method with B3-B5 mentioned in Fig. 7 due to the absence of existing methods, with the scores calculated on 10 editing results shown in Tab. 1. Please note that this metric might also inherit the bias from CLIP. We will keep finding better evaluation metrics and improve this evaluation in the future.

[1] Learning Neural Parametric Head Models.

[2] Extracting Training Data from Diffusion Models.

[3] Dreamface: Progressive generation of animatable 3d faces under text guidance.

[4] DeepFloyd IF. https://stability.ai/blog/deepfloyd-if-text-to-image-model.

[5] TextMesh: Generation of Realistic 3D Meshes From Text Prompts.

[6] ProlificDreamer: High-Fidelity and Diverse Text-to-3D Generation with Variational Score Distillation.

---

### Decision · Program_Chairs · 2023-09-21

**Decision:**

Accept (poster)

**Comment:**

All the reviewers are quite positive about this work. The rebuttal has addressed the concerns raised by the reviewer. This paper has presented a new approach to 3D avatar generation from pretrained text-to-image generation models. The generated 3D avatars are impressive, and the presented ideas are interesting and inspiring. The authors are recommended to incorporate experimental results in the rebuttal to the main paper or the supplement.

The authors are also recommended to take the suggested actions provided by the ethics reviewer.